# Relative Importance of Black Carbon, Brown Carbon and Absorption Enhancement from Clear Coatings in Biomass Burning Emissions

Rudra P. Pokhrel[1], Eric R. Beamesderfer[1, *], Nick L. Wagner[2], Justin M. Langridge[3], Daniel A. Lack[4], Thilina Jayarathne[5], Elizabeth A. Stone[5], Chelsea E. Stockwell[6], Robert J. Yokelson[6] and Shane M. Murphy[1]

[1]Department of Atmospheric Science, University of Wyoming, Laramie, Wyoming, USA
[2]NOAA Earth System Research Laboratory, Chemical Sciences Division, Boulder, Colorado, USA
[3]Observation Based Research, Met Office, Fitzroy Road, Exeter, EX1 3PB, UK
[4]Transport Emissions, Air Quality and Climate Consulting, Brisbane, Australia
[5]Department of Chemistry, University of Iowa, Iowa City, Iowa, USA
[6]Department of Chemistry, University of Montana, Missoula, Montana, USA
[*]Now at: School of Geography and Earth Science, McMaster University, Hamilton, Ontario, Canada

*Correspondence to*: Shane M. Murphy (shane.murphy@uwyo.edu)

**Abstract.** A wide range of globally significant biomass fuels were burned during the fourth Fire Lab at Missoula Experiment (FLAME-4). A multi-channel photoacoustic absorption spectrometer (PAS) measured dry absorption at 405, 532, and 660 nm and thermally denuded ($250^o$ C) absorption at 405 and 660 nm. Absorption coefficients were broken into contributions from black carbon (BC), brown carbon (BrC) and lensing following three different methodologies, with one extreme being a method that assumes the thermal denuder effectively removes organics and the other extreme being a method based on the assumption that black carbon (BC) has an angstrom exponent of unity. The methodologies employed provide ranges of potential importance of BrC to absorption but, on average, there was a factor of 2 difference in the ratio of the fraction of absorption attributable to brown carbon estimated by the two methods. BrC absorption at shorter visible wavelengths is of equal or greater importance to that of BC with maximum contributions of up to 92% of total aerosol absorption at 405 nm and up to 58% of total absorption at 532 nm. Lensing is estimated to contribute a maximum of 30% of total absorption, but typically contributes much less than this. Absorption enhancements and the estimated fraction of absorption from BrC show good correlation with the elemental to organic carbon ratio (EC/OC) of emitted aerosols and weaker correlation with the modified combustion efficiency (MCE). Previous studies have shown that brown carbon grows darker (larger imaginary refractive index) as the ratio of black to organic aerosol (OA) mass increases. This study is consistent with those findings but also demonstrates that the fraction of total absorption attributable to BrC shows the opposite trend: increasing as the organic fraction of aerosol emissions increases and the EC/OC ratio decreases.

## 1 Introduction

The significant impact of black carbon (BC) aerosol on radiative forcing is well established (Bond et al., 2013), but the magnitude and wavelength dependence of absorption by organic carbon (often called brown carbon, BrC) remains poorly constrained (Barnard et al, 2008; Lack et al., 2012a; Kirchstetter and Thatcher, 2012; Bahadur et al., 2012; McMeeking et al., 2014; Laskin et al., 2015; Olson et al., 2015). If BC is coated by non-absorbing organics (primary or secondary organic aerosol) or inorganic non-absorbing materials (ammonium sulfate, ammonium nitrate, etc.) these coatings can enhance the magnitude of absorption by the BC core. This effect is often called, somewhat inaccurately, lensing and for simplicity this term will be utilized in this paper (Fuller et al., 1999). The lensing effect is slightly decreased if the coatings themselves are absorbing (Lack and Cappa, 2010). While Mie calculations and lab studies support the notion that significant, 50-100%, increases in BC absorption can occur via lensing at atmospherically relevant aerosol sizes, observations of this effect in ambient air have yielded a wide range of results (Bond et al., 2006; Cappa et al., 2012; Nakayama et al., 2014). Studies have also found a wide range of possible imaginary refractive indices for BrC (Kirchstetter et al., 2004; Lack et al., 2012a; Alexander et al., 2008; Saleh et al., 2014), with Saleh et al. (2014) postulating that the imaginary refractive index can be predicted if the ratio of black carbon to organic aerosol mass (BC: OA) is known.

Potential sources of BrC include emissions from biomass burning (Kirchstetter et al., 2004; Moosmuller et al., 2009; Chen and Bond, 2010; Lack et al., 2012a; Saleh et al., 2014, Washenfelder et al., 2015), incomplete combustion of fossil fuels, especially coal (Bond, 2001; Yang et al., 2009; Olson et al., 2015), and secondary organic aerosols (Saleh et al., 2013; Zhang et al., 2013; Liu et al., 2014; Lin, et al., 2015). There exists significant uncertainty concerning the relative contribution of each of these source-types to total BrC concentrations, but several studies have identified biomass burning as a potentially significant source (Washenfelder et al., 2015; McMeeking et al., 2014; Lack et al., 2012a). Open biomass burning (BB) is one of the largest global sources of black carbon (BC) and organic carbon (OC) and biomass burning emissions have a significant direct effect on the Earth's radiative balance (Bond et al., 2013). When biomass burning emissions interact with clouds, there are significant semi-direct and indirect effects and the magnitude of the semi-direct effects depends on the optical properties of the emitted aerosols (Sakaeda et al., 2011; Lin et al., 2014; Jacobson 2014).

Absorption enhancement can be defined as the ratio of the absorption of ambient aerosol (with coatings present) to the absorption of BC alone. Experimental attempts to quantify absorption enhancement often employ thermal denuders to remove organic and inorganic coatings. Lack et al. (2012a) measured ambient aerosols with a large influence of biomass burning and found absorption enhancements up to 2.5 at 405 nm and 1.4 at 532 nm using a thermal denuder. Laboratory measurements of absorption enhancement in biomass burning aerosols show significant variation with values greater than 2 observed at 405 nm in some samples and values ranging from 1.2 – 1.5 at 532 and 781 nm (McMeeking et al., 2014). There are a number of studies that estimate the absorption due to BrC in the blue and ultraviolet wavelengths (Yuan et al., 2015; Washenfelder et al., 2015; Guo et al., 2014; Nakayama et al., 2014; Lack et al., 2012a; Cappa et al., 2012; Bahadur et al., 2012; Flowers et al., 2010; Wang et al., 2016; Stockwell et al., 2016). Results from these studies show large variations in the

estimated percentage of absorption attributable to BrC. Laboratory and field studies have measured wide ranges of BrC absorptivity values (Kirchstetter et al., 2004; Lack et al., 2012a; Alexander et al., 2008; Saleh et al., 2014). Recent model studies have shown that inclusion of BrC could significantly alter the direct radiative forcing due to carbonaceous aerosols (Saleh et al., 2015; Feng et al., 2013). Additionally, BrC is thought to lose its absorptivity in a relatively short (hours to days)

amount of time aging in the atmosphere (Wang et al., 2016; Forrister et al., 2015; Lee et al., 2014). This aging provides another reason it is critical to differentiate between absorption from BrC and BC.

To understand the relative importance of absorption from BrC versus lensing and BC, it is critical to understand the potential variability in attribution of absorption caused by different methodologies. Various studies show large variations in the percentage of absorption due to BrC (Yuan et al., 2015; Washenfelder et al., 2015; Guo et al., 2014; Nakayama et al., 2014;

Lack et al., 2012a; Cappa et al., 2012; Bahadur et al., 2012; Flowers et al., 2010) at blue wavelengths. While some of the variation is certainly due to variations in the ambient aerosol, some could potentially be the result of various approaches used to estimate the contribution of BrC to total light absorption in previous studies. Lack et al. (2012a) and Saleh et al. (2014) used core/shell Mie theory with inputs of BC core and shell size and the imaginary refractive index of BrC. Particle morphology is also taken in account, using the Rayleigh-Debye-Gans approximation, in some studies (Liu et al., 2015).

Others have used the difference in absorption enhancement from thermally denuding the aerosol at different wavelengths to determine the contribution from evaporated brown carbon (Guo et al., 2014; Nakayama et al., 2014; Cappa et al., 2012) with the assumptions that the lensing effect depends weakly on wavelength (Lack and Cappa, 2010). Another method that is simple and widely applied assumes an absorption Angstrom Exponent (AAE) value for BC (typically 1) and defines absorption above the predicted BC absorption at low wavelengths to be BrC, while sometimes also accounting for lensing

(Fialho et al., 2005; Favez et al., 2009; Bahadur et al., 2012; Cazorla et al., 2013; Yuan et al., 2015). In this study, we estimate the percentage of absorption due to BC, lensing, and BrC following methodologies based on assuming an AAE and based on thermally denuding the aerosol, thereby providing a range, that covers variations caused by experimental approach, of potential contributions to absorption in biomass burning emissions from BrC, lensing and BC.

Recent studies have shown that certain parameters can be used to parameterize biomass-burning aerosol optical properties.

These parameters include modified combustion efficiency (MCE), black carbon to organic aerosol ratio (BC/OA), and elemental to total carbon ratio ($\frac{EC}{EC+OC}$) (Pokhrel et al., 2016; Lu et al., 2015; Saleh et al., 2014; McMeeking et al., 2014; Liu et al., 2014). The current study demonstrates that the magnitude of total brown carbon absorption can be parameterized with the EC/OC ratio, a distinct result from Saleh et al. who demonstrated that the intensive quantity of absorptivity, or imaginary refractive index, can be parameterized with the BC/OA ratio.

**2 Materials and Methods**

Measurements were made during the multi-investigator FLAME-4 experiment. Details of the fuels burned and overall experiment can be found in Stockwell et al. (2014) and details of the experimental setup for optical measurements in Pokhrel

et al. (2016). The schematic of the instrumental setup during FLAME-4 experiment is shown in Fig. 1. Briefly, this study reports results from 12 different fuels with significant global emissions over 22 individual burns, as detailed in Figure 4. All results presented are from burns where a fire was ignited in a large combustion room at the Missoula Fire Sciences Laboratory and allowed to burn to completion. Measurements were made after smoldering had ceased and emissions from all stages of the fire had become well mixed in the combustion room.

## 2.1 Inlet System

Aerosol was pulled from an inlet placed roughly 3 meters above the floor of a (12.5 m × 12.5 m × 22 m) combustion room once the smoke was well mixed (typically 15-20 minutes after the start of a burn). Fuels burned include ponderosa pine, black spruce, rice straw, organic hay, organic and conventional wheat, sugarcane, giant cut sawgrass, wiregrass, African grass, chamise, manzanita, and North Carolina peat. Smoke was transferred to the instruments at 10 L min$^{-1}$ through a 1/2 inch OD copper line. Aerosol was passed through a cyclone impactor that removed particles with aerodynamic diameters larger than 2.5 µm and diluted with particle-free air in order to maintain extinction levels near 500 Mm$^{-1}$ to avoid saturation of sampling instrumentation. It is important to note that aerosol sampled for EC/OC measurements (Section 2.7) was not diluted. Next, the air was dried by two 100-tube Nafion driers (Perma Pure, Toms River, NJ) operated in parallel, which reduced the relative humidity in the sample cell to less than 15 %. Following the Nafion driers, an activated carbon monolith (MAST Carbon, Basingstoke, UK) was used to scrub $NO_x$ and $O_3$ from the sample air while transmitting the particles. The successful removal of $NO_x$ was continuously tracked by a cavity ring down spectrometer (CRDS) gas-phase channel at 405 nm. A filter was periodically inserted (in an interval of 5 to 10 minutes) into the sample stream to remove particles and confirm baseline stability.

## 2.2 Instrumentation

Aerosol absorption measurements were made with a multi-wavelength photoacoustic absorption spectrometer (PAS) instrument (Lack et al., 2006; 2012b). The PAS was configured with five cells that measured absorption coefficients of dry aerosol (RH < 15%) at 405, 532, and 660 nm and denuded aerosol at 405 and 660 nm. An integral component of the PAS was a thermal denuder system. The thermal denuder deployed during FLAME-4 was a 50 cm long, 0.43" ID stainless steel tube heated to 250°C followed by a 50 cm 0.43" ID unheated tube with an activated carbon honeycomb monolith (MAST Carbon, Basingstoke, UK). The denuder was run at 250$^0$C in an attempt to avoid charring of organic aerosol which would artificially increase absorbing material in the denuded channel. The flow rate through the thermal denuder was 5 lpm. The activated carbon section absorbed volatile species that evaporate from the aerosol in the heated region. This denuder configuration was shown to have almost complete removal (from what core) of a semi-volatile organic compound (Di-Ethyl-Hexyl-Sebacate having boiling point of 300 $^0$C) coating sodium chloride cores (Fierz et al., 2007). It is known that biomass burning emissions often have low volatility material and these materials can be absorbing (May et al., 2013; Saleh et al., 2014). It is possible, in fact probable given that the denuder was run at 250$^0$C to avoid charring, that all of the absorbing

organic material was not removed by the thermal denuder in this study, which will be discussed in detail in the results and discussion.

## 2.3 PAS Calibration

A Cavity Ring-down Spectrometer (CRDS) (Langridge et al., 2011) that operates at identical wavelengths to the PAS (660 nm, 532 nm, 405 nm) was used to calibrate PAS absorption measurements. Calibrations were conducted using ozone sampled simultaneously by the PAS and CRDS. PAS absorption at a given wavelength was set equal to the measured extinction from the CRDS at that same wavelength, with Raleigh scattering subtracted.  Further details of the PAS calibration using the CRDS can be found in Lack et al. (2012b). Further details about the instrumental setup of the PAS and CRDS can be found in Pokhrel et al. (2016). The uncertainty associated with the PAS measurements during FLAME-4 was found to be 7% (1σ) in the 660 and 532 nm channels and 18% (1σ) in the 405 nm channel (Pokhrel et al., 2016). The higher uncertainty in the 405 nm channel is due to the use of ozone during PAS calibration. The absorption cross section of ozone at 405 nm is $1.51\times10^{-23}$ cm$^2$/ molecule (Axson et al., 2011), which is two orders of magnitude less than the absorption cross section in the 450-858 nm range ($10^{-21}$ cm$^2$/molecule) (Anderson and Mauersberger, 1992). During ozone calibration, the absorption coefficients at 405 nm were always less than 20 Mm$^{-1}$, which resulted in a calibration curve with significantly more uncertainty than the curves for the other wavelengths when extrapolated to the high absorption values observed in this study.

## 2.4 Particles loss in Thermal denuder

Particle losses occur inside the thermal denuder because of thermophoretic and diffusional processes (Wehner et al., 2002). Particle loss in the thermal denuder was measured by passing glassy carbon spheres through the denuder at a flow rate of 5 lpm. Different sized particles were passed through the thermal denuder at multiple temperatures. Figure 2 shows the percentage of particles that were transmitted through the thermal denuder. The percentage of particles lost in the thermal denuder depends on both particle size and thermal denuder temperature. During FLAME-4, the vast majority of particles for all burns were less than 300 nm (Saleh et al., 2014) and the temperature of the thermal denuder was set to 250°C. Under these conditions, the transmission efficiency of the thermal denuder is 71% on average.  Based on this, denuded absorption was divided by 0.71 to correct for losses. It is important to note that this size range and temperature result in nearly the maximal particle loss through the denuder observed in Figure 2 and it is therefore possible that particle losses through the denuder could be overcorrected.  If this occurs, it will lead to an underestimate of absorption enhancement in particles. The efficiency curves are relatively insensitive to diameter making it a reasonable assumption to have a fixed efficiency factor and not have to consider complications associated with particle size, and thus transmission efficiency, changing as the particle evaporates on passing through the denuder. Absorption enhancements based on the thermal denuder measurements as low as 0.91 were observed during the Flame-IV study, suggesting denuder losses were overcorrected by 10% in some

cases. Because this error is larger than what would be expected based on Figure 2, we used these in-situ observations to justify the application of a ± 10% uncertainty to our denuder-derived absorption enhancements due to uncertainty in particle losses (see Section 2.5 for details).

Because absorption enhancements derived from thermally denuded channels in this study were obtained by comparing denuded and dry channels at the same wavelength run in parallel, it was critical to ensure that both the dry and denuded channels at a given wavelength gave nearly identical absorption coefficients when the denuder was not inline. To verify this, numerous thermal denuder bypasses were completed with the purpose of ensuring that enhanced absorption observed in dry channels was caused by coatings and not by instrumental drift of other issues. During a thermal denuder bypass, inlet air was diverted around the thermal denuder and directly to the cell that normally measured denuded aerosol. Several thermal denuder bypasses were done during each day and the absorption coefficients measured by the dry and denuded channels were compared when measuring identical input aerosol. Figure 3 below shows a comparison of absorption coefficients for dry and denuded channels measured during thermal denuder bypasses. For 660 nm the slope between the absorption coefficients measured by the dry and denuded channels is within 3% of unity. However, at 405 nm, the two channels differ by as much as 52% (a day with one of the largest observed differences is shown). This difference is largely due to the relatively large (18%, 1σ) error in the calibration of the 405 nm channel, but is still at the outer limits of the Gaussian error curve created when the errors are added in quadrature. The idea that the large differences between the channels are more than random error is supported by the observation that the dry channel was found to be consistently higher than the denuded channel. The very high correlation (R above 0.99) between the denuded and dry channels gives confidence that this is an error in the absolute calibrations of the channels and that the magnitude of the instrument response through the day is consistent between the two channels. The reason for the denuded 405 nm channel having consistently lower signal than the dry 405 nm channel remains unclear. However, during the last 3 days of experimentation new critical orifices were added to the ozone calibration system allowing the introduction of higher levels of ozone. The calibration of the dry 405 nm channel determined without the high ozone points (what was done for most of the project) was consistently closer to the slope determined using all ozone concentrations (including the high ozone points) than the calibration of the denuded 405 nm channel without the high-ozone points. These results suggest that the dry channel calibrations were more accurate than the denuded channel calibrations and, because of this the denuded channel absorption was adjusted to match the dry channel absorption. The adjustment factors for each day of measurement are provided in Table S1.

## 2.5 Absorption Enhancement and Absorption Angstrom Exponent

Absorption enhancement is defined here as the amount of absorption observed when all material is present (BC and coatings) vs. the absorption observed when just BC is present. One way to estimate absorption enhancement is to measure the absorption of particles that have been thermally denuded at a high enough temperature to remove organics versus the absorption from unperturbed particles (Lack et al., 2012a)

$$E_{AbsDen} = \frac{b_{abs\_dry(\lambda)}}{b_{abs\_den(\lambda)}} \qquad (1)$$

where $b_{abs\_dry(\lambda)}$ is the absorption coefficient of ambient particles at a specific wavelength and $b_{abs\_den(\lambda)}$ is the absorption coefficient at the same wavelength after the particles are heated in a thermal denuder. The thermal denuder in this experiment was run at $250^\circ$ C. The error in the $E_{AbsDen}$ calculated in this way does not depend on the absolute accuracy of either the dry or the denuded absorption because denuded absorption was adjusted to match the dry absorption during thermal denuder bypasses and while the corrections can be large they were very stable throughout a given day (Figure 3). Instead, the dominant errors in this calculation are the possibility that the denuder does not remove all organics and the need to adjust for particle losses in the denuder. The minimum value of $E_{AbsDen}$ calculated in this study was ~0.91, observed during measurements of a fire with a very high modified combustion efficiency and presumably very little non-BC material being emitted. This result is only possible if particle loss in the thermal denuder is overcorrected and based on this we derive the error in $E_{AbsDen}$ due to particle loss corrections to be $\pm 10$ % and add this uncertainty in quadrature to one standard deviation of the average measured value to report the total uncertainty in the $E_{AbsDen}$. This total error does not account for the potential error of not fully removing organics in the denuder, since this error cannot be quantified. AAE is defined as $b_{abs} = a \, \lambda^{-AAE}$ where $b_{abs}$ is the absorption coefficient and the constant, a, is independent of wavelength. AAE is estimated from the slope of least square fit of the logarithm of absorption coefficients vs. the logarithm of wavelengths. AAE are determined from the three wavelengths (405, 532, and 660 nm).

## 2.6 Modified Combustion Efficiency (MCE)

The modified combustion efficiency is defined as (Ward and Radke, 1993; Yokelson et al., 1997)

$$MCE = \frac{\Delta CO_2}{\Delta CO + \Delta CO_2} \qquad (2)$$

Where $\Delta CO$ and $\Delta CO_2$ are the mixing ratio enhancements above background. Background mixing ratios were measured before the ignition of each burn. The CO and $CO_2$ mixing ratios were measured by an open path Fourier transform infrared spectrometer (Stockwell et al., 2014). The MCE reported in this study is the fire-integrated value.

## 2.7 Determination of Elemental Carbon to Organic Carbon Ratio (EC/OC)

EC/OC estimates were made in identical fashion to Pokhrel et al. (2016), but are described here again for clarity. Fine particulate matter (PM2.5) was selected by a cyclone operating at a flow rate of 42 L min−1 and was collected on to 37 mm quartz fiber filters (QFF; PALL, Port Washington, NY) at ambient temperature. Field blanks were collected at a rate of one in seven samples. Prior to use, QFF were pre-cleaned by baking at 550 ∘C for 18 h. Filters were stored in cleaned aluminum-foil-lined Petri dishes sealed with Teflon tape, and stored frozen (−20 ∘C) before and after analysis. OC and EC were measured by thermal optical analysis (Sunset Laboratories, Forest Grove, OR, USA) following the IMPROVE-A protocol where the EC/ OC split was determined by thermal optical transmittance. The effects of positive sampling artifacts due to carbonaceous gas adsorption were assessed using quartz filters behind Teflon (QBT; Cheng et al., 2009) for 14 of the

96 fires, including grass, rice straw, ponderosa pine, black spruce, and peat. For fires with QBT collected, the OC on the backup filter was subtracted directly. For fires without backup filters or those that were below the detection limit, the average OC correction for that fuel type was applied: rice straw (2.0 ± 0.4 %), ponderosa pine (1.2 %), black spruce (2.9 ± 1.6 %), and peat (3.1 ± 0.8 %). For fuel types without backup filters collected, the study average OC artifact (2.4 ± 1.2 %) was subtracted. This approach to artifact correction assumes that the amount of carbonaceous gas adsorbed is proportional to the mass concentration of OC; this assumption is considered to be reasonable because the back-up filters contained less than 5.6 µg OC cm$^{-2}$ and similar quartz fiber filters become saturated above 6 µg OC cm$^{-2}$ (Turpin et al., 1994). Analytical uncertainties for OC were propagated from the standard deviation of field blanks (0.7 µg cm$^{-2}$) and 5 % of the OC concentration. For EC, uncertainties were propagated from an estimate of the instrument precision (0.1 µg m$^{-2}$), 5 % of EC concentration and 5 % of pyrolyzed carbon (which forms from OC charring on the filter during analysis). The value of 5 % is a conservative estimate of the precision error in replicate sample analysis, which is typically 1–3 % (NIOSH, 1999). Analytical uncertainties for the EC/ OC ratio were propagated from the individual EC and OC uncertainties.

## 3 Results and Discussion

### 3.1 Absorption Enhancement Derived with a Thermal Denuder

Absorption enhancement determined by comparing aerosol passed through a thermal denuder at 250$^0$C to non-thermally-denuded aerosol (E$_{AbsDen}$, Eq. 1) was calculated during 22 individual burns of twelve different fuels. The E$_{AbsDen}$ values reported in this study are the average value obtained over approximately one hour of measurements that were made after the smoke had completely mixed in the combustion room. Figure 4 shows bar plots of the average E$_{AbsDen}$ for different fuels. Repeated burns of the same fuel often generated different burn conditions resulting in different MCE (Eq. 2) which are also given. The total bar height (red plus blue) represents the absorption enhancement at 405 nm and red bars represent E$_{absDen}$ at 660 nm. Fuels are categorized into four different groups: coniferous trees, crop residues, grasses and brushwood, and peat. These fuel categories have large contributions to the total biomass burning across different parts of world (Page et al, 2002; Chang et al., 2010; Clinton et al., 2006; McCarty et al., 2007). For all burns, E$_{AbsDen}$ is larger at 405 nm than at 660 nm, except for a Giant Saw Grass (GSG) burn where the two are nearly identical. In this burn, both the 405 and 660 nm E$_{AbsDen}$ are unity within experimental uncertainty, suggesting the BC emitted during this burn had very little coating. The fact that all burns, except giant saw grass and wire grass, have an E$_{AbsDen}$ at 405 nm significantly larger than that at 660 nm provides evidence of the presence of BrC on most of the burns since the lensing effect typically has a weak dependence on wavelength (Lack and Cappa, 2010; McMeeking et al., 2014).

E$_{AbsDen}$ at 660 nm ranges from 0.92 ± 0.09 to 1.43 ± 0.17, similar to the range of E$_{AbsDen}$ observed during Flame-3 for a different suite of fuels. Flame-3 E$_{AbsDen}$ results at 532 and 781 nm ranged from 1.2-1.5 (McMeeking et al., 2014) but the thermal denuder in that study was operated at or below 150 $^0$C while the thermal denuder in this study was run at 250 $^0$C. Our results at 660 nm are also similar to the E$_{Abs}$ of 1.4 measured at 532 nm in a biomass plume near Boulder, CO (Lack et

al., 2012a). The range of $E_{AbsDen}$ at 405 nm observed in this study is similar to previous studies (McMeeking et al., 2014; Lack et al., 2012a) except for peat where much higher absorption enhancement is observed. The peat burns give a very high value (5.65 ± 1.43) of $E_{AbsDen}$ at 405 nm because smoldering emissions from peat are predominantly BrC with a negligible amount BC content (Chakrabarty et al., 2016; Pokhrel et al., 2016; Stockwell et al., 2016). It is evident from Fig. 4 that $E_{AbsDen}$ can vary significantly, depending on burn conditions, even for the same fuel.

## 3.2 Parameterization of Absorption Enhancement

In order to gain a better understanding of what drives variation in $E_{AbsDen}$, we examined correlations between absorption enhancement and other fire-relevant variables. It is notable that some regressions are done for a semi-log plot while others are linear or log-log. The type of regression was chosen based on objective criterion for simple regression. Namely that the residuals are equally scattered from the regression line and that the residuals are as close as possible to a normal distribution. The model (either LogY vs LogX, logY vs X, or Y vs LogX) which satisfied these criterion for simple linear regression was chosen. Fuel type alone is insufficient because $E_{AbsDen}$ varies dramatically during different burns of a single fuel. Figure 5 shows $E_{AbsDen}$ at 405 nm versus the absorption Angstrom exponent, elemental to organic carbon ratio (EC/OC), and modified combustion efficiency (MCE, Eq. 2). AAE values in Figure 5 were calculated from a best fit to the logarithm of absorption coefficient versus wavelength at three wavelengths (660 nm, 532 nm, 405 nm) as detailed in Pokhrel et al. (2016). A strong positive correlation (r = 0.96) between absorption enhancement at 405 nm and AAE is observed for all fuels. The linear relationship between logarithm of $E_{AbsDen}$ and AAE suggests that brown carbon absorption can be parameterized within the AAE framework. While the exact nature of the trend is notable, the trend itself is expected because the AAE of pure BC is typically near 1 (Kirchstetter and Thatcher, 2012; Wiegand et al., 2014) and anything larger than that strongly suggests the presence of absorbing coatings.

Fig. 5(a) shows that for AAE less than 2, $E_{AbsDen}$ at 405 nm remains close to 1, indicating little influence from BrC or coatings when emissions have a low AAE. Also for AAE less than 2, $E_{AbsDen}$ at 660 nm (Figure 6(a)) are similar to $E_{AbsDen}$ at 405 nm, again indicating little presence of brown carbon, which is expected to absorb mainly at shorter wavelengths. Although $E_{AbsDen}$ at 405 nm shows strong correlation with AAE, $E_{AbsDen}$ at 660 nm does not show good correlation, with a Pearson's correlation coefficient for logarithm of $E_{AbsDen}$ vs. AAE of 0.96 at 405 nm but only 0.32 at 660 nm. This result strongly suggests that either brown carbon is not present at 660 nm or, if there is brown carbon at 660 nm, it is not strongly correlated to AAE. This means that, if $E_{AbsDen}$ 660 nm is purely from lensing, the effect of non-absorbing coatings on absorption cannot be easily parameterized with AAE.

Figures 4 (b) and 5 (b) show that $E_{AbsDen}$ at both 405 and 660 nm decrease linearly (r = -0.89 and -0.78 respectively) with the logarithm of the EC/OC ratio. When aerosol composition has more EC than OC, $E_{AbsDen}$ at 405 and 660 nm both approach 1 as the effect of lensing and BrC become minimal. The slope of the 660 and 405 nm $E_{absDen}$ vs. EC/OC fits are very different and as the fraction of OC increases, $E_{AbsDen}$ at 405 nm grows much more quickly than $E_{AbsDen}$ at 660 nm. Saleh et al. (2014) estimate that the average BC core and particle size during Flame-IV was 100 and 200 nm respectively. In these size ranges,

core-shell Mie theory predicts that the lensing effect will be similar at 660 and 405 nm (McMeeking et al., 2014). Given this, the much larger $E_{AbsDen}$ at 405 nm indicates absorption from BrC. A key observation is that $E_{AbsDen}$ at 405 nm can be parameterized with EC/OC without the need to explicitly define fuel type. There is relatively poor correlation between $E_{AbsDen}$ at either 405 or 660 nm with MCE. There are fewer data points for EC/OC parameterizations because not all burns

measured with the optical suite had corresponding EC/OC measurements. Accordingly, the linear fit of $E_{AbsDen}$ vs. MCE at 405 nm does not include peat and one ponderosa pine burn (with MCE~0.83) because there was not EC/OC data for these burns and they represented significantly lower MCE than all other burns.  For similar MCE values, $E_{AbsDen}$ at 405 nm varies by a factor of 3 in some burns. A potential reason for the poor fit with MCE is the difficulty of MCE to predict aerosol properties such as BC/OA or EC/OC (Grieshop et al., 2009; Pokhrel et al., 2016) on which absorptivity of organic aerosol

has a strong dependence (Saleh et al., 2014).

### 3.3 Contribution to Total Absorption from Brown Carbon and Lensing

The dataset collected during this study allows us to estimate the contribution due to BC, BrC, and lensing in several different, commonly implemented, ways as discussed in the introduction.  Here we describe the specific approaches used and evaluate the difference in the results generated by these approaches. A conceptual representation of these approaches is

shown in Fig. 7.

**Approach 1: Assume The Thermal Denuder Removes all Organic Carbon and Assume $E_{Abs}$ From Lensing is Constant at All Wavelengths.**

In this approach $E_{AbsDen}$ at 660 nm is assumed as an enhancement due to lensing. We assume the absorption enhancement from lensing is identical at 405 nm and 660 nm (Guo et al., 2014; Nakayama et al., 2014; Cappa et al., 2012). We assume the

absorption of the denuded channel at 405 nm is due entirely to BC. The remaining $E_{AbsDen}$ after lensing is subtracted is assumed to be caused by brown carbon. The following equations summarize these assumptions and describe how we derive the absorption from brown carbon at 405 nm.

$$b_{abs\_405\_BrC} = b_{abs\_405\_dry} - E_{abs\_660} \times b_{abs\_405\_den} \tag{3}$$

where $b_{abs\_405\_BrC}$ is absorption due to BrC at 405 nm, $b_{abs\_405\_dry}$ is non-denuded dry absorption measured at 405 nm,

$b_{abs\_405\_den}$ is denuded absorption measured at 405 nm, and $E_{abs\_660}$ is absorption enhancement at 660 nm. Due to incomplete removal of low-volatile organics, estimated fraction of absorption due to BrC using approach 1 is most likely underestimated. The logic is that Eq. (3) can be simplified as

$$b_{abs\_405\_BrC} = b_{abs\_405\_dry} - E_{abs\_660} \times b_{abs\_405\_den}$$

$$b_{abs\_405\_BrC} = b_{abs\_405\_dry} - \frac{b_{abs\_660\_dry}}{b_{abs\_660\_den}} \times b_{abs\_405\_den}$$

The denuded absorption at both 660 and 405 nm will be overestimated due to incomplete removal of organics, but the problem is expected to be worse at 405 nm because both brown carbon and lensing increase the 405 nm denuded absorption while lensing is the dominant effect for the 660 denuded absorption. Given this, the ratio $\frac{b_{abs\_405\_den}}{b_{abs\_660\_den}}$ is expected to be larger than one and hence BrC absorption will be underestimated because both the dry absorptions (405, 660 nm) will not be affected. To extend our results to 532 nm, where dry, but not denuded, absorption was measured by the PAS we calculated the absorption due to BrC as follows:

$$b_{abs\_532\_Brc} = b_{abs\_532\_dry} - E_{abs\_660} \times b_{abs\_405\_den} \times \left(\frac{405}{532}\right)^{AAE_{den}} \tag{4}$$

where $b_{abs\_532\_Brc}$ is absorption due to BrC at 532 nm, $b_{abs\_532\_dry}$ is absorption measured at dry phase at 532 nm, $E_{abs\_660}$ is absorption enhancement at 660 nm, and $AAE_{den}$ is the absorption angstrom exponent calculated between the 660 nm and 405 nm denuded channels. The BrC absorption estimated by equations 3 and 4 can be visualized by subtracting dry absorption and the line labelled "BC + Lens (1)" in Fig. 7.

**Approach 2: Assume 660 nm Denuded Absorption Represents BC Absorption, Assume the AAE of BC is 1, and Assume E$_{Abs}$ From Lensing is Constant at All Wavelengths.**

In this approach we assume the absorption measured in the denuded 660 nm channel is due entirely to BC. We estimate absorption due to BC at wavelengths less than 660 nm using an AAE of 1 for BC. This approach may be more accurate than approach 1 if the thermal denuder does not effectively remove all of the brown carbon absorption at 405 nm. This approach will be incorrect if the AAE of BC is different than 1, if there is brown carbon at 660 nm that is not removed by the denuder or if the denuder generates BC at 660 nm. E$_{AbsDen}$ at 660 nm is considered to be due to lensing of the BC core and same lensing effect is applied on all wavelengths. Absorption due to BC at any wavelength is estimated by using equation (5) (line "BC (2,3)" in Fig. 7)

$$b_{abs\_\lambda_1\_BC} = b_{abs\_660\_den} \times \left(\frac{660}{\lambda_1}\right)^{1} \tag{5}$$

and absorption due to BrC is estimated by:

$$b_{abs\_\lambda_1\_BrC} = b_{abs\_\lambda_1\_dry} - E_{abs\_660} \times b_{abs\_\lambda_1\_BC} \tag{6}$$

Substituting the value of $b_{abs\_\lambda_1\_BC}$ from equation (5) generates an alternate expression for absorption due to BrC:

$$b_{abs\_\lambda_1\_BrC} = b_{abs\_\lambda_1\_dry} - E_{abs\_660} \times b_{abs\_660\_den} \times \left(\frac{660}{\lambda_1}\right)^{1} \tag{7}$$

where $\lambda_1$ is the desired wavelength, which in this study is 405 and 532 nm. Absorption from lensing is calculated as the difference between total absorption and the sum of the contributions from BC and BrC to absorption. BrC absorption estimation using equation 7 can be visualized by subtracting line "BC + Lens (2)" from the dry absorption in Fig. 7.

**Alternate Description of Approach 2**

A widely used approach is to assume that absorption at 660 nm or higher wavelengths is equivalent to BC absorption plus lensing from the clear coating because there is no absorption contribution from BrC. It is also assumed that coated BC particles have an AAE of 1, similar to uncoated BC. Absorption due to BC at lower wavelengths is estimated as

$$b_{abs\_\lambda_1\_BC} = b_{abs\_660\_dry} \times \left(\frac{660}{\lambda_1}\right)^1 \tag{8}$$

And absorption due to BrC is estimated as

$$b_{abs\_\lambda_1\_BrC} = b_{abs\_\lambda_1\_dry} - b_{abs\_660\_dry} \times \left(\frac{660}{\lambda_1}\right)^1 \tag{9}$$

Interestingly, because $b_{abs\_660\_dry} = E_{abs\_660} * b_{abs\_660\_den}$, equation (9) and equation (7) are equivalent. Therefore, this approach gives the same absorption due to BrC as what was described for approach 2 and there is no need to call this a separate approach. It is described here because it is widely implemented by groups without a thermal denuder and to demonstrate that several different assumptions lead to the same numerical result for BrC absorption. Importantly, equations (8) and (9) do not allow for assessment of the impact of lensing while equations (5) and (6) do. A final note is that one also arrives at equation (9) if it is assumed that lensing has a negligible impact on BC properties and all absorption at 660 nm is from BC. However, this is a rather unrealistic approach given that lensing has clearly been shown to impact absorption at 660 nm.

**Approach 3: Assume 660 nm Denuded is BC Absorption, Assume 660 nm Dry is BC Plus Lensing, Assume Clear-Coated BC has AAE of 1.6.**

AAE for non-denuded BC has been commonly assigned as 1.0 in many past studies, including some recent studies (Kirchstetter and Thatcher, 2012; Wiegand et al., 2014). However, it has been shown theoretically that with a non-absorbing coating, AAE of BC can be as large as 1.6 (Gyawali et al., 2009; Lack and Cappa, 2010). Here we use AAE of 1 for uncoated BC and AAE of 1.6 for coated BC. This approach estimates an approximate maximum increase in lensing with decreasing wavelength rather than assuming it is constant with wavelength as in the other approaches. It is an important reference point because while BrC may bleach (Wang et al., 2016; Forrister et al., 2015; Lee et al., 2014), absorption enhancement from lensing will remain unless coatings evaporate. Absorption due to BC, lensing, and BrC is calculated as follows:

Absorption due to BC is estimated as

$$b_{abs\_\lambda_1\_BC} = b_{abs\_660\_den} \times \left(\frac{660}{\lambda_1}\right)^1 \tag{10}$$

And absorption due to BrC is estimated by

$$b_{abs\_\lambda_1\_BrC} = b_{abs\_\lambda_1\_dry} - b_{abs\_660\_dry} \times \left(\frac{660}{\lambda_1}\right)^{1.6} \tag{11}$$

where $b_{abs\_\lambda_1\_BC}$ is absorption due to BC at wavelength $\lambda_1$, $b_{abs\_660\_den}$ is denuded absorption measured at 660 nm, $b_{abs\_\lambda_1\_BrC}$ is absorption due to BrC at $\lambda_1$, $b_{abs\_\lambda_1\_dry}$ is non-denuded absorption measured at $\lambda_1$, $b_{abs\_660\_dry}$ is non-denuded absorption measured at 660 nm, and $\lambda_1$ is the desired wavelength (405 and 532 nm in this study). BrC estimated by equation

11 can be visualized as the difference between the dry absorption and line "BC + Lens (3)" in Fig. 7. Absorption contribution due to lensing is estimated by subtracting absorption due to BC and BrC from the total absorption.

Based on these approaches, the contribution of BC, BrC, and lensing to total absorption by biomass-burning aerosol under different burn conditions is estimated. Table 1 summarizes the results at 405 nm. BrC at 405 nm is estimated to contribute 0 to 92 percentage of total biomass-burning aerosol absorption depending upon the burn and the approach used. From multiple burns of the same fuel (pine, Cali. rice straw, black spruce, or. hay, or. wheat) it is evident that the BrC contribution to absorption changes significantly for a single fuel demonstrating the importance of burn conditions. Between the three different approaches, the lower bound for the contribution of BrC estimation is consistently approach 1 and the upper bound is consistently approach 2, with approach 3 being in the middle. We hypothesize that the reason for approach 1 consistently resulting in the lowest fraction of absorption from BrC is incomplete removal of organics by the thermal denuder, an idea that is supported by the observation that the AAE of the denuded aerosol channels (405 and 660 nm) was often significantly larger than one, which is often observed for uncoated BC (average $AAE_{den}$ = 1.9).  It is thought that the resonance time of aerosol in the thermal denuder may have been insufficient to remove the significant coatings present in some burns or that the temperature ($250^0$C) could have been too low to remove extremely low volatility organic compounds that have been suggested to be important contributors to brown carbon (Saleh et al., 2014). While this suggests that the BrC contributions derived from approach 2 may be closer to reality, approach 1 provides a useful lower bound to BrC absorption and even this lower bound is often a significant fraction of total absorption. The difference in BrC contribution predicted by approach 1 and 2 varies from burn to burn with a maximum ratio of 4.3 and a mean ratio of 2.1, demonstrating the significant variation between methodologies and the potential difficulties of assessing BrC via thermal denuding. Similar to the 405 nm results, the BrC contribution at 532 nm is important to overall absorption and shows large variations, ranging from 0 to 58 percent as shown in Table S2. The percentage of absorption due to BrC at 532 nm also has a strong dependency on fuel type and the methodology used, similar to the 405 nm results.

The maximum ratio of the BrC contributions found in approach 1 and 2 is 2.5 at 532 nm, lower than the maximum difference at 405 nm. However, the mean ratio is 2.1, which is identical to the 405 nm results. We find that coatings can contribute up to roughly 30% of the absorption, but generally the contribution due to coating is low relative to BC and BrC. However, it should be noted that incomplete removal of organics by the thermal denuder would result in an underestimate of absorption enhancement from clear coatings (lensing) and an overestimate of the relative importance of BC in all three approaches. Approach 3 consistently yields the highest contribution from coatings, suggesting that an angstrom exponent of 1.6 for BC with clear coatings is indeed at the high end of possible AAE values.

Our estimation of BrC contribution to total biomass burning aerosol absorption shows large variations, but it is also clear that, independent of methodology, the contribution of BrC to total absorption is large and often of similar or greater magnitude than the contribution from BC. Most ambient studies show relatively low (<20%) BrC contribution to absorption at 405 nm (Yuan et al., 2015; Nakayama et al., 2014; Cappa et al., 2012), but these studies were not focused on biomass burning emissions and had significantly less biomass burning influence. Lack et al.(2012a) estimated that BrC contributed 27

± 15% of absorption at 405 nm during a wild fire in Boulder, Colorado based on thermal denuder measurements. Additional studies focused on biomass burning reported ranges of values for the contribution to absorption from BrC (McMeeking et al., 2014; Flowers et al., 2009) and these ranges are similar to ours. Flowers et al. (2009) showed BrC can contribute 27-51% of absorption at 405 nm using literature mass absorption cross section (MAC) values to estimate the absorption due to BC and

calculating absorption due to BrC by comparing measured absorption with estimated absorption based on MAC and BC mass. McMeeking et al. (2014) found, based on fresh emissions from a wide range of biomass fuels, that nonrefractory particulate matter can contribute 20-80% of the light absorption at wavelengths ≤532 nm, a result similar to the range of contributions to absorption from brown carbon (0-92%) found in this study.

### 3.4 Parameterization of Brown Carbon Absorption

Given the large variation in the contribution of BrC to total absorption for different fuels and for repeated burns of the same fuel, it is clear that some type of framework for estimating the significance of brown carbon absorption is needed. Saleh et al. (2014) developed a methodology to estimate the imaginary part of the refractive index of brown carbon based on the ratio of black carbon to organic aerosol mass. This approach requires detailed particle size measurements that were not available in the current study. Instead, we directly parameterize the contribution of BrC to biomass burning aerosol absorption with

MCE, EC/OC, and AAE. Figure 8 shows the percentage of absorption due to BrC at 405 and 532 nm estimated with approaches 1, 2 and 3 vs EC/OC and AAE. The general pattern is that the fraction of absorption due to BrC is highest when the aerosol composition is dominated by OC. On the other hand, aerosol composition dominated by EC show a lower contribution to total absorption from BrC at both wavelengths. Generally, the trend of the fraction of BrC absorption vs. EC/OC is similar for different approaches, but the slopes of the lines change significantly based on the approach. The

percentage of absorption due to BrC at 405 and 532 nm shows a reasonably good correlation with EC/OC as demonstrated by the correlation coefficients between 0.7 and 0.9 shown in the Figure 8 (a), (b), and (c). We also parameterize the BrC percentage with AAE and find that AAE shows very good correlation with the percentage of absorption due to BrC yielding correlation coefficients between 0.8 and 0.99. Fig. 8 (d, e, and f) shows that for an AAE less than 2, contributions from BrC are less than about 20% and decrease with decreasing AAE regardless of the approach used to estimate these values. Also,

there is a higher contribution to absorption from BrC at 405 vs 532 nm at AAEs above roughly 2.5. On the other hand, MCE (Fig. S1) does not have an easily fitted relationship with the percentage of absorption from BrC at either wavelength. For an MCE of 0.92 or greater, the percentage of absorption by BrC varies by a factor of 3 to 4 for very similar MCE values. The coefficients for each fitted line in Fig. 8 are reported in Table S3.

### 4 Conclusions

In this study 12 different fuels, representing globally significant sources of biomass burning aerosol emissions, were burned in 22 individual burns. Absorption enhancements determined based on thermally denuding the aerosol ($E_{AbsDen}$) at 405 nm

are large when the biomass plume is dominated by organic carbon emissions. We observed large variations in $E_{AbsDen}$ at 405 nm with a maximum value of $5.6 \pm 1.0$. $E_{AbsDen}$ at 660 nm is generally low (with the maximum value of $1.433 \pm 0.166$) suggesting that lensing is a less important contributor to biomass burning aerosol absorption than brown carbon (BrC) at blue end of the visible spectrum. From multiple burns of the same fuel, it is observed that $E_{AbsDen}$ has a strong dependency on burn

condition and that knowledge of fuel type is insufficient to predict BrC. $E_{AbsDen}$ shows good correlation with both AAE (r = 0.96) and EC/OC (r = -0.89) but has limited correlation with MCE (r=-0.39). We suspect that the $E_{AbsDen}$ values at 660 and 405 nm are lower bounds because the thermal denuder may not have removed all organic material. To address this concern, we estimated the fraction of total aerosol absorption due to BC, BrC, and lensing and found that the fraction of absorption from BrC could vary by up to a factor of 4.3 depending on the approach utilized. The average ratio of the fraction of

absorption from brown carbon found via a method assuming the AAE of BC was 1 versus the thermal denuder method is 2.1 at 405 nm. We found that BrC can contribute up to 92% of total aerosol absorption at 405 nm and up to 58% of total absorption at 532 nm while lensing can contribute a maximum of 30% but typically contributes much less than this, though the impact of lensing could be underestimated if the denuder did not remove all organic material. The fraction of absorption from BrC shows reasonably good correlation with AAE and EC/OC at both 405 and 532 nm (independent of the approach

used) but has a poor correlation with MCE. The fractional absorption due to BrC at 405 and 532 nm increases as the EC/OC ratio decreases. This result is distinct but not inconsistent with Saleh et al. (2014) who found that the imaginary index of refraction increases with increasing BC/OA ratio. These two results can be understood with the idea that brown carbon grows darker as emissions have a higher fraction of black carbon relative to non-refractory organic mass, but the fraction of total absorption caused by brown carbon increases as the amount of organic mass increases and the black carbon to organic

carbon mass ratio decreases.

## 5 Data availability

All the data presented in this paper can be accessed via e-mail request to Shane Murphy ([shane.murphy@uwyo.edu](mailto:shane.murphy@uwyo.edu)).

*Acknowledgements.* This material is based upon work supported by the National Science Foundation under grant no.

1241479. Chelsea E. Stockwell and Robert J. Yokelson were supported primarily by NSF grant ATM-0936321. Thilina Jayarathne and Elizabeth A. Stone were supported by the University of Iowa. We thank Ted Christian, Dorothy L. Fibiger, and Shunsuke Nakao for assistance with filter sample collection and sample preparation. We appreciate the contribution of Eric Miller, DavidWeise, Greg Askins, Guenter Engling, Savitri Garivait, Christian L'Orange, Benjamin Legendre, Brian Jenkins, Emily Lincoln, Navashni Govender, Chris Geron, and Kary Peterson for harvesting the fuels for this study.

Collection of Indonesian peat by Kevin Ryan and Mark Cochrane was supported by NASA Earth Science Division Award NX13AP46. We also thank Daniel Murphy for valuable suggestions during data collection and manuscript preparation.

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

Table 1: Percentage of absorption due to BC, lensing (clear coating), and BrC from biomass burning aerosol emissions at 405 nm estimated from three different approaches. The ID is the fire ID assigned during FLAME-4 for particular burns. The ratio in the rightmost column is the ratio in BrC absorption estimated by approach 2 vs. approach 1. Results where the BrC contribution was found to be zero by approach 1 are not assigned a ratio. NA indicates not available.

| ID | Materials | Approach 1 | | | Approach 2 | | | Approach 3 | | | Ratio |
|---|---|---|---|---|---|---|---|---|---|---|---|
| | | BC | Coat | BrC | BC | Coat | BrC | BC | Coat | BrC | |
| 129 | Pine | 38 | 3 | 58 | 14 | 1 | 85 | 14 | 6 | 80 | 1.5 |
| 142 | Pine | 68 | 11 | 21 | 48 | 10 | 42 | 48 | 30 | 22 | 2.0 |
| 144 | Pine | 55 | 21 | 23 | 30 | 14 | 57 | 30 | 29 | 42 | 2.5 |
| 130 | California rice straw | 64 | 2 | 34 | 37 | 2 | 62 | 37 | 15 | 49 | 1.8 |
| 143 | California rice straw | 35 | 4 | 61 | 16 | 2 | 82 | 16 | 8 | 76 | 1.3 |
| 131 | Black Spruce | 59 | 2 | 39 | 37 | 2 | 60 | 37 | 16 | 46 | 1.5 |
| 134 | Black Spruce | 55 | 10 | 35 | 35 | 6 | 59 | 35 | 20 | 45 | 1.7 |
| 138 | Organic Hay | NA | NA | NA | 29 | 11 | 60 | 29 | 24 | 47 | NA |
| 146 | Organic Hay | 41 | 7 | 53 | 28 | 0 | 72 | 29 | 9 | 62 | 1.4 |
| 132 | Organic Wheat | 70 | 1 | 29 | 48 | 1 | 52 | 48 | 17 | 35 | 1.8 |
| 149 | Organic Wheat | 73 | 11 | 16 | 40 | 6 | 54 | 40 | 21 | 38 | 3.4 |
| 139 | Giant saw grass | 100 | 0 | 0 | 86 | 0 | 14 | 86 | 14 | 0 | - |
| 148 | Giant saw grass | NA | NA | NA | NA | NA | NA | 66 | 22 | 12 | NA |
| 133 | Conventional Wheat | 68 | 1 | 32 | 45 | 1 | 54 | 45 | 17 | 38 | 1.7 |
| 135 | Chamise | 93 | 0 | 7 | 64 | 0 | 37 | 64 | 21 | 15 | 5.3 |
| 136 | Manzanita | 92 | 0 | 8 | 73 | 0 | 28 | 73 | 24 | 3 | 3.5 |
| 141 | Wire grass | 100 | 0 | 0 | 100 | 0 | 0 | 100 | 0 | 0 | - |

| 147 | Sugar cane | 40 | 5 | 55 | 21 | 2 | 77 | 21 | 10 | 69 | 1.4 |
| 150 | NC peat | 16 | 2 | 82 | 8 | 0 | 92 | 8 | 3 | 89 | 1.1 |

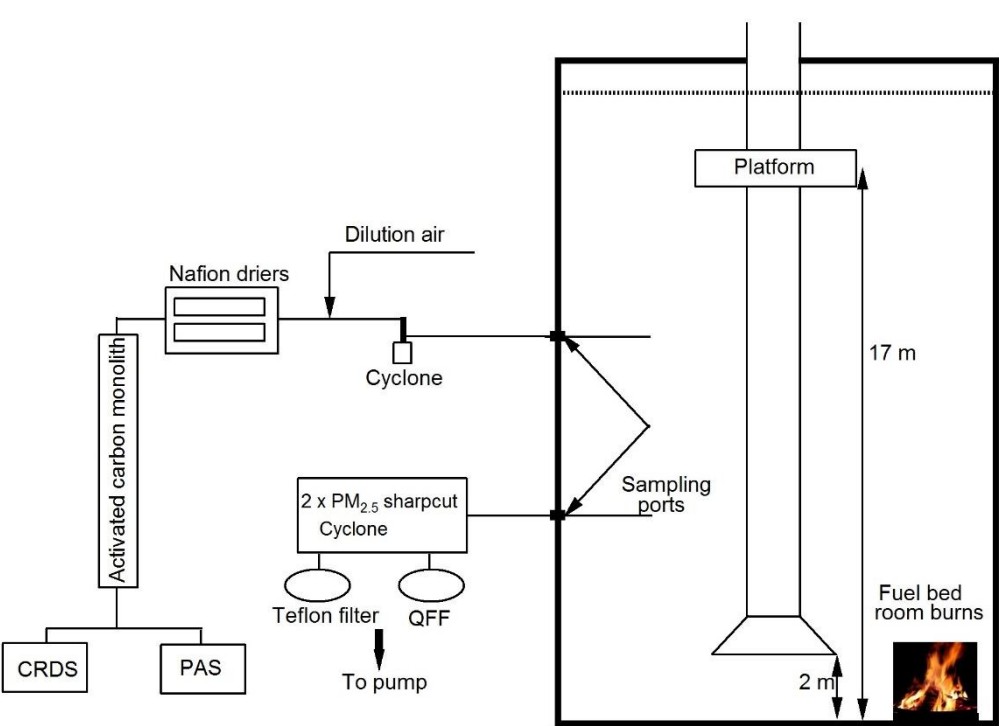

Figure 1: The schematic of instrumental setup during FLAME-4 experiment.

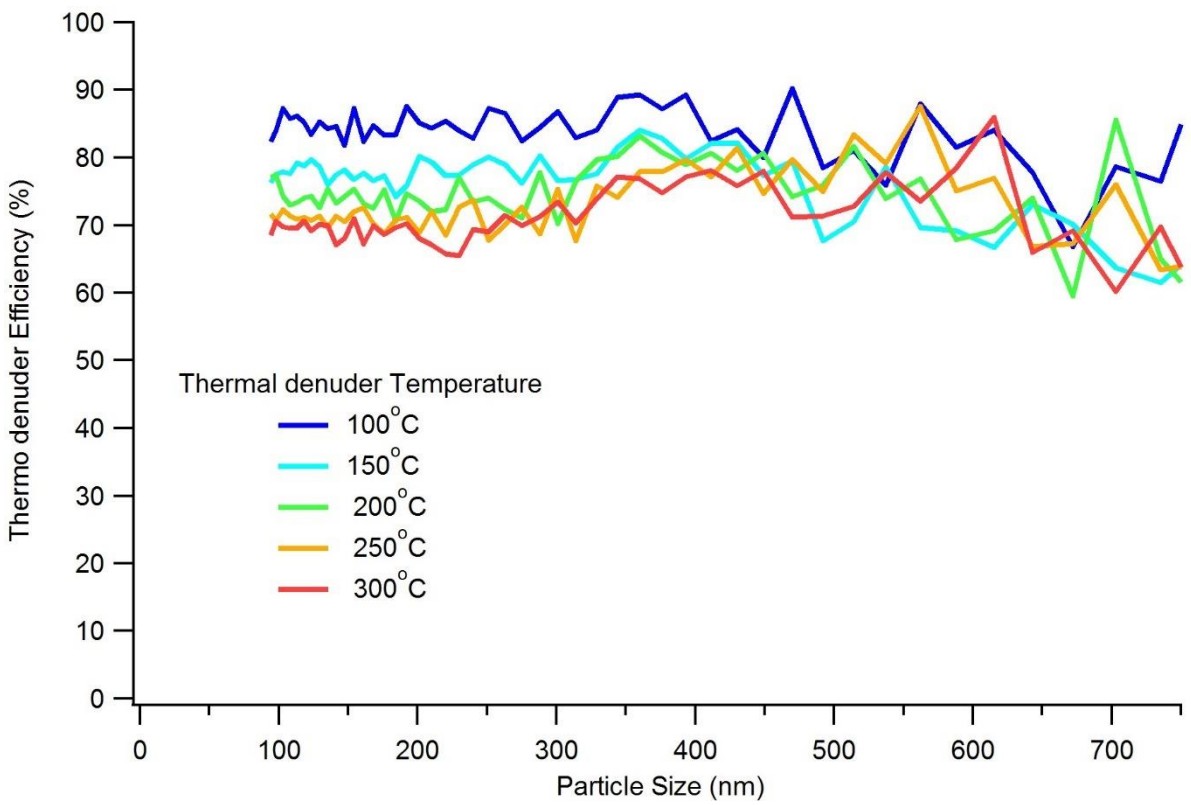

Figure 2: Particles loss through the thermal denuder as a function of particle size and thermal denuder temperature.

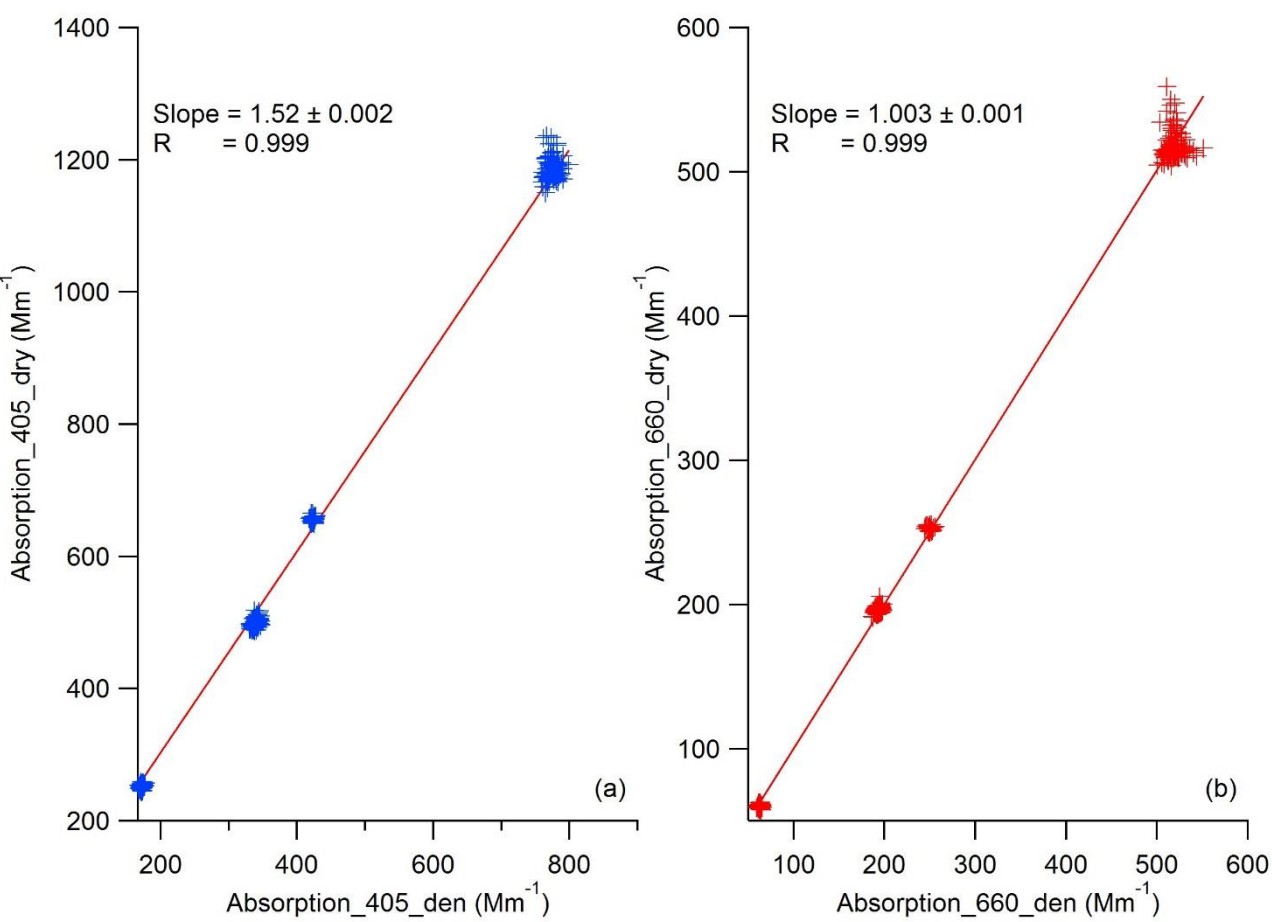

Figure 3: Comparison of absorption coefficients measured by dry and denuded channels during thermal denuder bypasses for a day with significant differences in the 405 nm channel. (a) 405 nm (b) 660 nm.

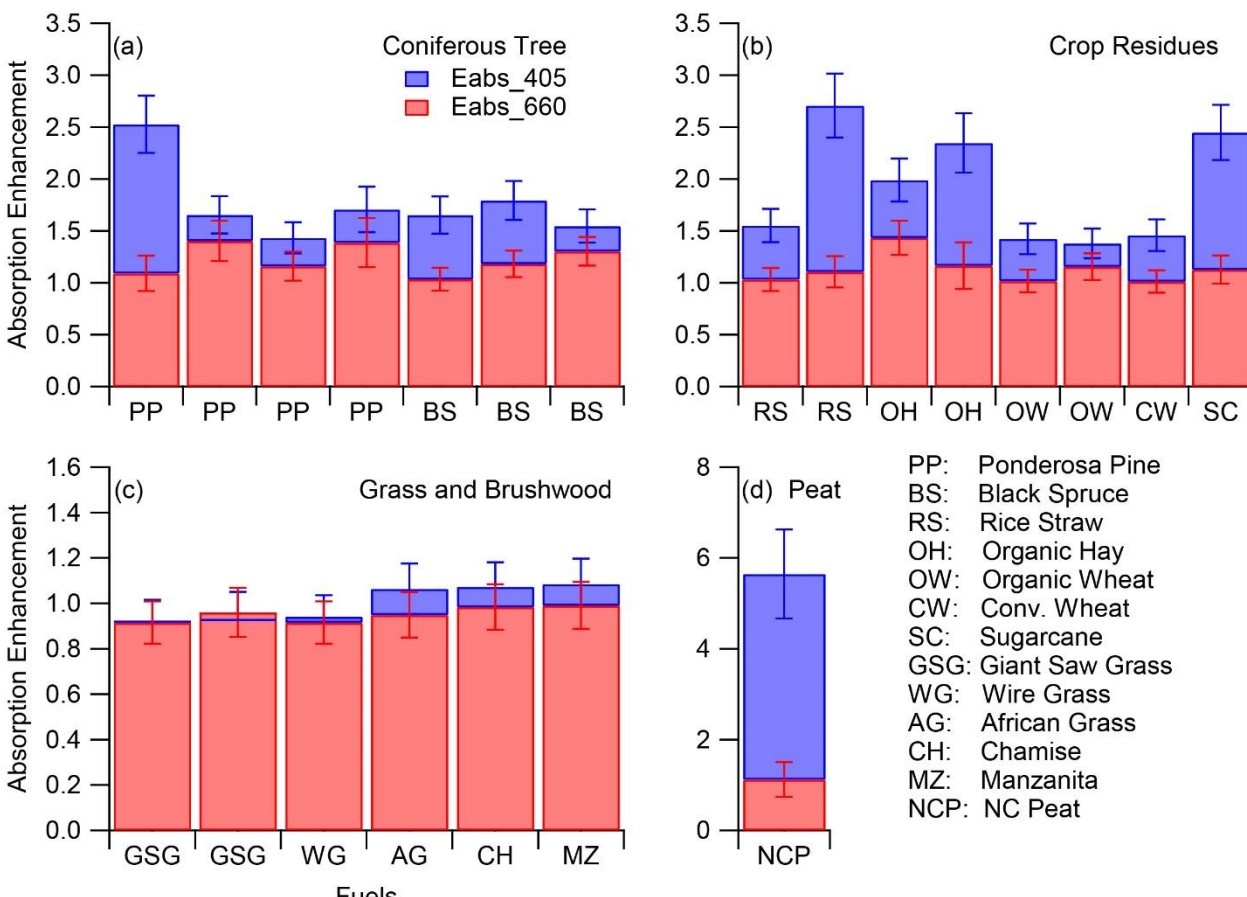

Figure 4: Bar plots of absorption enhancements derived by thermally denuding aerosol and measuring denuded and non-denuded particles at both 405 and 660 nm. Results are grouped in terms of fuel types. Total bar heights (red + blue) are representative of absorption enhancement at 405 nm while red bars represent absorption enhancement at 660 nm. Figure (a) is for coniferous trees, (b) is for crop residues, (c) is for grass and brushwood and (d) is for peat. The legend shows the name of each fuels reported in each group.

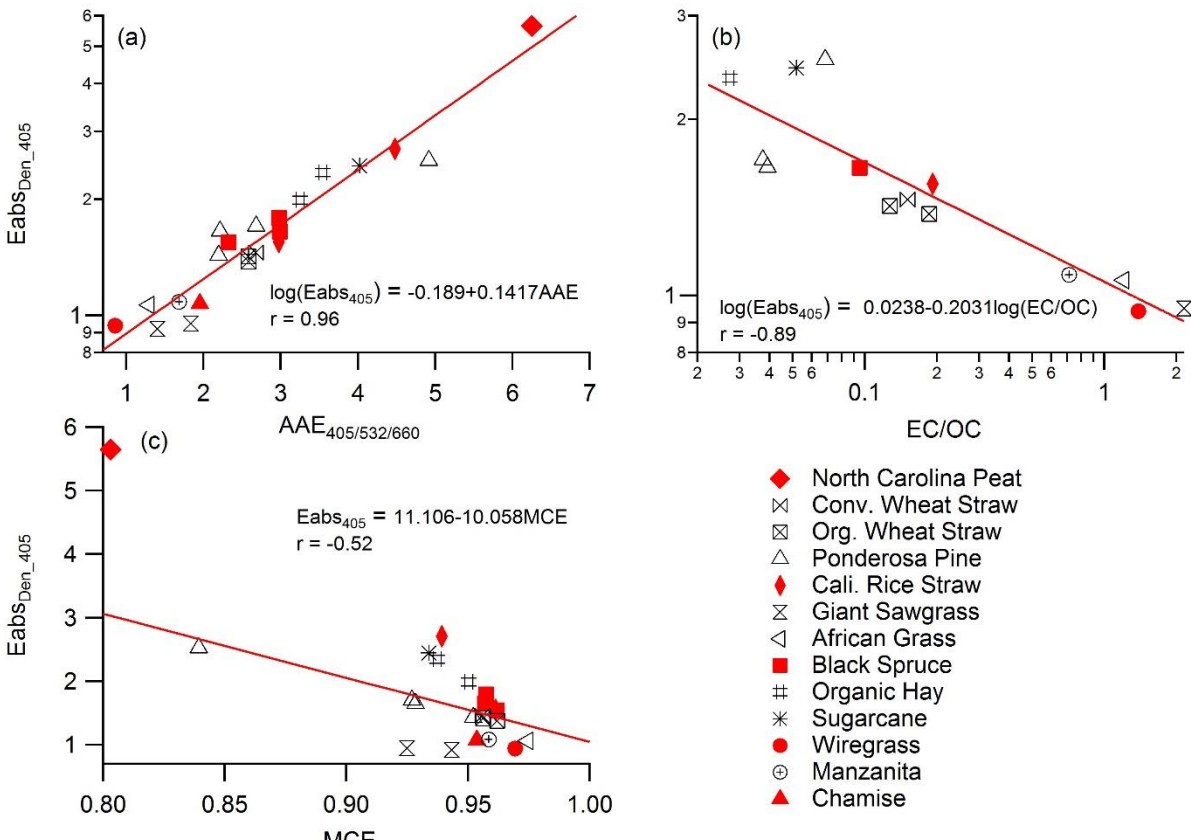

Figure 5: 405 nm absorption enhancement parameterized with (a) AAE, (b) EC/OC, and (c) MCE. The symbols are sorted by fuel type and listed in the legend. Red lines are least squares fits of the data with the equation and correlation coefficient (r) reported for each case. Given the near absence of BC, peat burns are considered as outliers and not included in model fit.

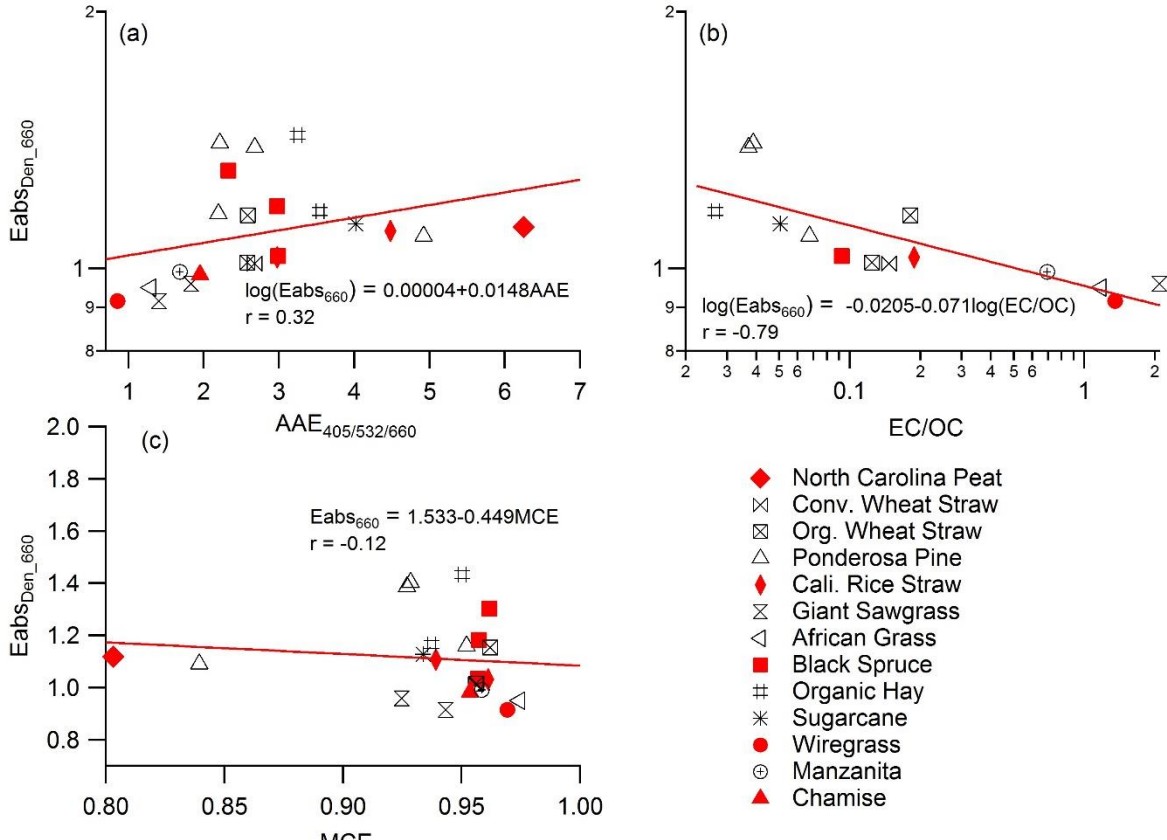

Figure 6: Same as Figure 5 but for 660 nm.

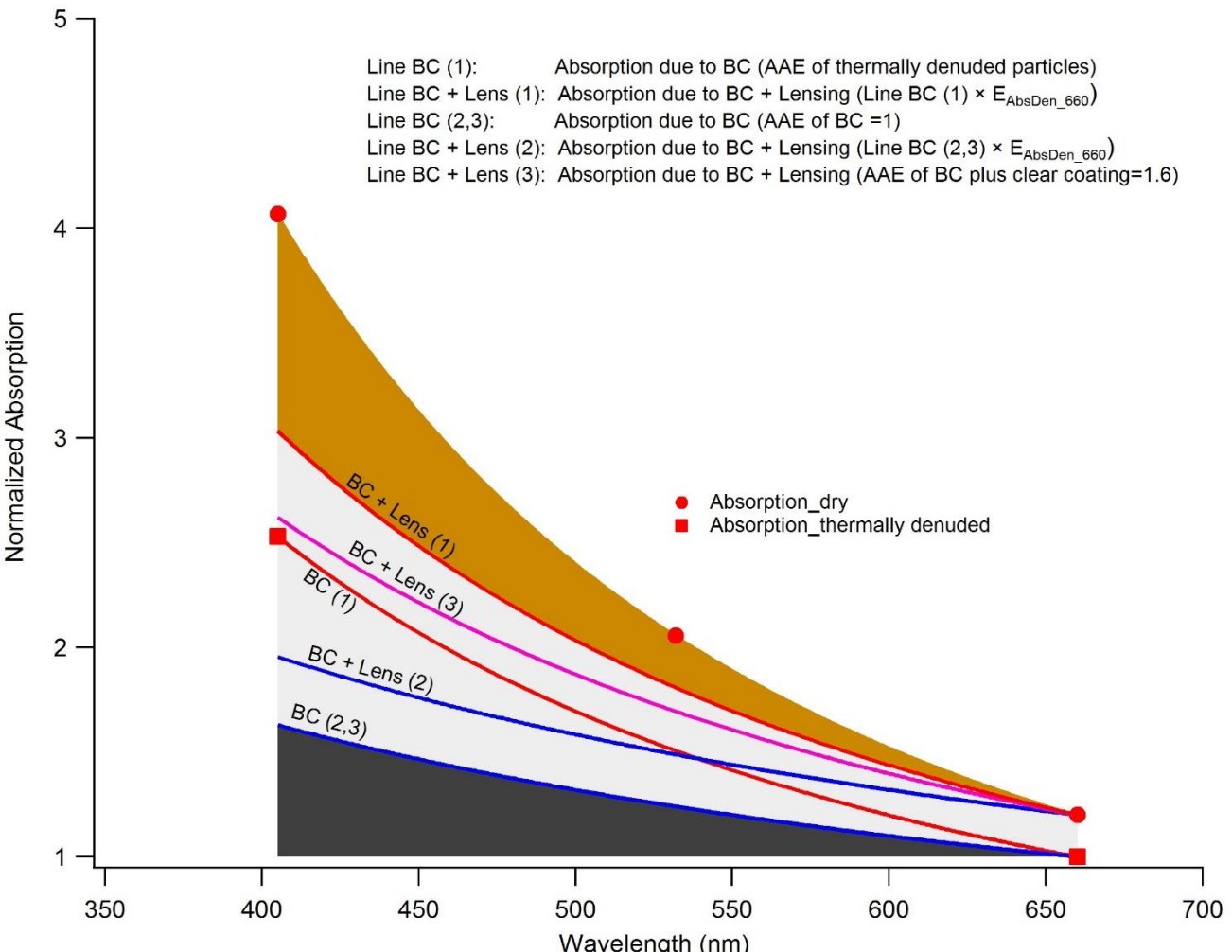

Figure 7: Conceptual representation of the three different approaches used to estimate the fraction of total absorption contributed by BC, BrC, and lensing. Numbers in parentheses correspond to the approach represented by that line. The y-axis is normalized so that the denuded absorption of the 660 nm channel is unity.

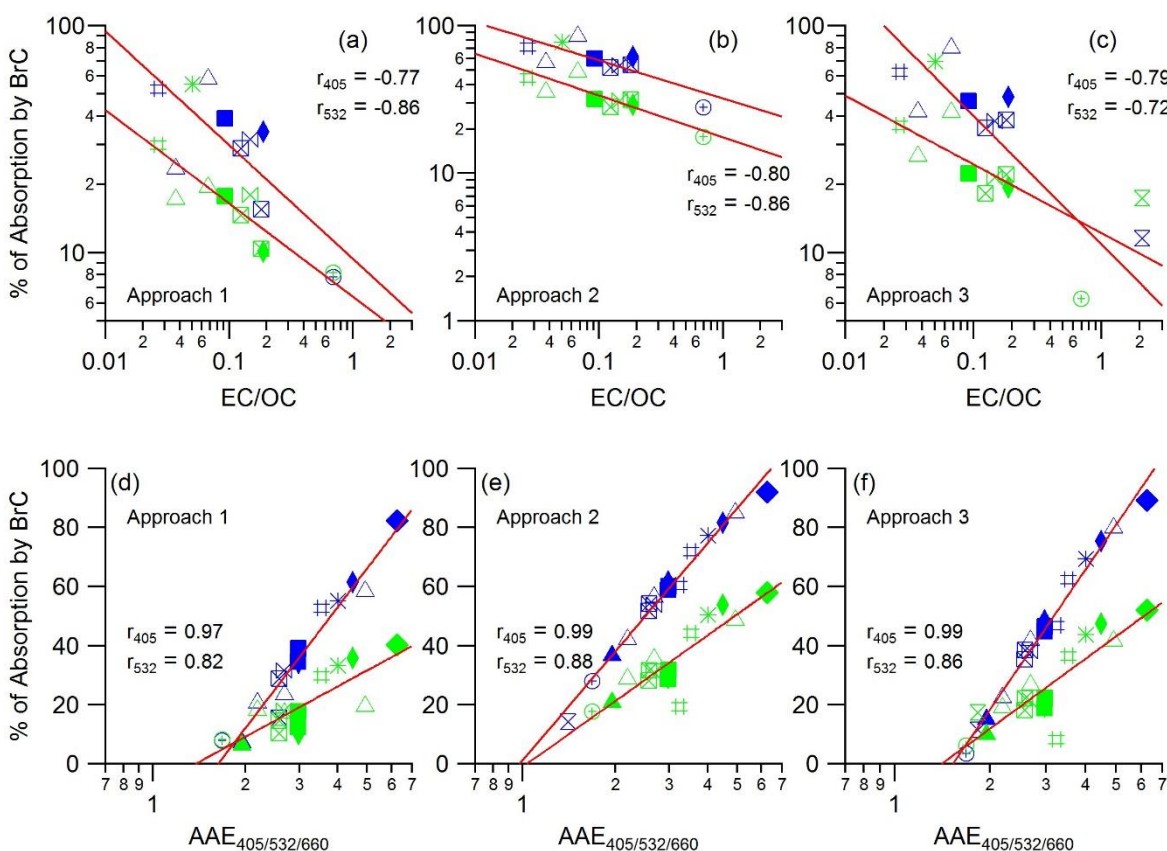

Figure 8: Percentage of absorption due to BrC vs. EC/OC (Figure (a) for approach 1, (b) for approach 2, and (c) for approach 3). Blue symbols are for 405 nm and green symbols are for 532 nm. Figure (d), (e), and (f) are the same as the top row of data except vs. AAE.

