# Peer review of "Relative Importance of Black Carbon, Brown Carbon and Absorption Enhancement from Clear Coatings in Biomass Burning Emissions"

_Atmospheric Chemistry and Physics, 2016_

## Referee Comment (RC1) · Anonymous Referee #1 · 20 Dec 2016

The authors used different methodologies to quantify the relative contributions of absorption from BC, BrC and lensing for 12 different fuels with significant global emissions over 22 individual burns. They demonstrated that the BrC was an important contributor to biomass burning aerosol absorption at blue end of the visible spectrum. The work is original, contain important addition to existing literature. The paper is clearly written, and is suited for publication to ACP. However, there are several concerns that should be addressed or considered before being accepted for publication.

Major comments:

(1) In this work, three different methodologies were used to calculate the contributions of absorption from BC, BrC and lensing. The authors descripted the methodologies in

the Results and Discussion section (Sec. 3.3). I would suggest the three approaches discussed in the Sec. 2 (Materials and Method).

(2) I would suggest providing a schematic of the instrument setup in the Materials and Method section.

(3) The efficiency of the thermal removal mechanism of non-refractory component is a critical point, but in this work is poorly investigated. Incomplete removal of organics (e.g. low-volatile BrC coatings on BC surface) by the thermal denuder would influence the estimate in the relative contributions from BC, BrC and lensing, especially for approach 1. It will be helpful to estimate the removal of low-volatile component after the heating stage. If the author cannot quantify the efficiency of the thermal removal, they can make a sensitive analysis to discuss the uncertainties of relative contributions from BC, rBC and lensing due to incomplete removal of low-volatile organics.

(4) Due to incomplete removal of low-volatile organics, the Eabs\_660 was underestimated and the babs\_405\_den was overestimated in Eq. (3), which would result in an unclear uncertainty (overestimate, underestimate or counteraction) in the fraction of absorption from BrC using approach 1. Meanwhile, the assumption of coated BC with an AAE of 1 in approach 2 led to an overestimate in the fraction of absorption from BrC. I do not know why the author could concluded that the BrC contributions derived from approach 2 was closer to reality than approach 1.

Specific points:

(1) P. 1 L 17: The temperate unit should be " $^{\circ}C$  " .

(2) P. 1 L 18-20: Three approaches was used to calculate the contributions of absorption from BC, BrC and lensing. However, the author only described two methodologies (i.e. with one.....and the other....).

(3) P. 7 L 12-14: Please define the absorption Angstrom exponent (AAE), such as using an equation.

---

## Referee Comment (RC2) · Anonymous Referee #2 · 27 Dec 2016

This manuscript presents a study of relative role of BC, BrC and absorption enhancement from clear coatings in biomass burning emissions. Various types of biomass fuels which representing globally significant sources of biomass burning aerosol emissions were burned in 22 individual burns in the experiment introduced in this work. The manuscript is outlined logically and written clearly. The results provide some insights into the relative importance of BC, BrC and lensing effect to total absorption coefficients in biomass burning emissions. This work may help for parameterization of BrC absorption used in models. However, more in-depth discussions are needed in the text before jumping into quantitatively conclusions. I suggest that his manuscript can be accepted for publication in this journal after my following comments are addressed.

Major comments:

1. Several assumptions are used when estimating the contributions of BC, BrC and lensing effect to total absorption coefficients. One of the assumptions is that the absorption enhancement from lensing is constant at all wavelengths, and this assumption is used in Approach 1 and 2. Another assumption is that clear coated BC has an AAE of 1.6, and this assumption is used in approach 3. According to references introduced by authors, these two assumptions are based on Mie calculation when all BC are core-shell mixed with other aerosol components. However, the mixing state of BC in this study is unknown. Obviously, the mixing state of BC will exert significant influences on the calculated contributions of BC, BrC and lensing effect. More analysis and discussions are needed to address the possible mixing the state of BC. I suggest that the authors estimate the influences of mixing state based on calculations in this study.

2. Results of parallel runs of thermally denuded channel and dry channel demonstrate that significant discrepancy exist (larger than 50 % which is at the outer limits of the Gaussian error curce) between measured absorption coefficients at 405 nm from two different photoacoustic absorption spectrometers. Though results of added experiments show that the dry 405 channel is better and absorption coefficient of the denuded channel is corrected according results of parallel runs, large bias still exist in measured absorption coefficient at 405 nm and this error will propagate into the calculation of AAE and influences the analytical results introduced in this research. Hence, uncertainty analysis about calculated AAE should be given in the text to acknowledge the awareness of this problem.

Specific comments:

3. Title of table 1, " at 405 nm estimated from four different approaches", it should be three approaches.

4. Caption of Figure 2, the unit of absorption coefficient should be included.

---

## Referee Comment (RC3) · Anonymous Referee #3 · 28 Dec 2016

This paper reports the relative contributions of light absorption of BC, lensing effect, and BrC for biomass burning aerosol from several types of fuels. The experiments were carefully conducted and manuscript is clearly written. Especially, the comparison of three different approaches to estimate the relative contributions of light absorption of BC, lensing effect, and BrC should be valuable for atmospheric science community. However, some descriptions in the experimental section and discussion on the points below are not enough. I recommend publication to ACP after the points below have been addressed.

Major comments:

1) For the correlation analyses between Eabs and AAE, EC/OC ratio, or MCE and

between the fraction of BrC absorption and AAE, EC/OC ratio, or MCE, the authors sometimes use logarithm but not in some other cases. For example, the author reported that the log(Eabs\_Den\_405) linearly correlates with AAE (Fig. 4), and the "% of absorption of BrC" linearly correlates with log(AAE) (not AAE itself) (Fig. 7). In contrast, the log-log plot was used for both of Eabs vs. EC/OC and % of absorption of BrC vs. EC/OC. The authors need to add detailed explanation on these choices. For the discussion of these correlations, the term "logarithm of" should be added in the text, if the correlation analysis was conducted for the logarithm.

2) This paper reported the small lensing effect for all fuels and burning conditions. Is the magnitude of the lensing effect reasonable, if you assume all OC is used for the coating with the core-shell structure?

3) The main findings of the paper "The fraction of absorption from BrC shows reasonably good correlation with AAE and EC/OC at both 405 and 532 nm." are not surprising, because the relative contribution of BrC is expected to increase with increasing OC concentration and that the AAE is expected to increase with increasing BrC (OC). The results imply that the light absorbing properties (such as mass absorption cross section and imaginary part of refractive index) of OC do not largely change with EC/OC ratio, fuel types, and burning conditions. If so, the results obtained in this work may be inconsistent with the results of Saleh et al. (2014). I think that the discussion on this point and the more detailed descriptions on the evidence leading to the conclusion below should be added. "This result is distinct but not inconsistent with Saleh et al. (2014) who found that the imaginary index of refraction increases with increasing BC/OA ratio. These two results can be understood with the idea that brown carbon grows darker as emissions have a higher fraction of black carbon relative to non-refractory organic mass, but the fraction of total absorption caused by brown carbon increases as the amount of organic mass increases and the black carbon to organic carbon mass ratio decreases."

Specific comments:
1) Page 3, 2 Material and Methods If a part of the data (fuel) used in this work is same with that used in Pokhrel (ACP2016), it is better to add the information on that and how the authors chose the data (fuel) used in this work.

2) Page 4, 2.1 Inlet system

a)Have you estimate the particle losses in the Nafion dryer and the activated carbon monolith?

b)Did you check the possible contribution of removal of semi-volatile organic compounds to the amount and optical properties of BrC through the change in the gas/particle partitioning?

c)Line 17: How often did you insert the filter for the baseline measurements?

3) Page 5, 2.4 Particle loss in Thermal denuder

a)How did you measure the particle size dependence of particle loss? Did you use two SMPS system and place them upstream and downstream of thermal denuder?

b)In Fig. 1, the particle transmittance only above 100 nm were given. How was the transmittance for smaller particles? If the particles with a diameter less than 100 nm are negligible, I recommend to the authors to give some information on this point.

4) Page 6, lines 22-24 "The calibration of the dry 405 nm channel determined without the high ozone points (what was done for most of the project) was consistently closer to the slope determined using all ozone concentrations (including the high ozone points) than the calibration of the denuded 405 nm channel without the high-ozone points." => I recommend to adding more qualitative information on this point.

5) Page 7, line 31 "For fires without backup filters or those that were below the detection limit, the average OC correction for that fuel type was applied: rice straw ( $2.0 \pm 0.4 \%$ ), ponderosa pine (1.2 %), black spruce ( $2.9 \pm 1.6 \%$ ), and peat ( $3.1 \pm 0.8 \%$ ). For fuel types without backup filters collected, the study average OC artifact ( $2.4 \pm 1.2 \%$ ) was

**ACPD**
subtracted." => It seems that the authors assume the amount of carbonaceous gas adsorption is proportional to the mass concentration of OC. The assumption may not be reasonable if the filter is saturated.

6) Page 9, line 24 Are the particle sizes (100 and 200 nm) given here volume-based average diameters or number based average diameters?

7) Page 14, line 2 "Figure 5" may be "Figure 7"

---

## Short Comment (SC1) · 9 Jan 2017

The authors used different methods to investigate the relative contributions of BC, BrC, and coating enhancement to the total absorption of biomass burning aerosols. The analysis and results could improve our understanding on aerosol absorption for biomass burning emissions. I have a short comment.

Recent studies showed that BC optical properties are also significantly influenced by particle (coating) structures in addition to coating thickness (He et al., 2015, 2016), which could be an important uncertainty source in determining aerosol absorption. It would be useful if the author could include these references and add some discussions on this aspect to highlight potential uncertainty associated with these important factors

in affecting BC/BrC absorption.

References:

He, C., Liou, K.-N., Takano, Y., Zhang, R., Levy Zamora, M., Yang, P., Li, Q., and Leung, L. R.: Variation of the radiative properties during black carbon aging: theoretical and experimental intercomparison, Atmos. Chem. Phys., 15, 11967-11980, doi:10.5194/acp-15-11967-2015, 2015.

He, C., Y. Takano, K. N. Liou, P. Yang, Q. B. Li, and D. W. Mackowski: Intercomparison of the GOS approach, superposition T-matrix method, and laboratory measurements for black carbon optical properties during aging, J. Quant. Spectrosc. Radiat. Transf., 184, 287–296, doi:10.1016/j.jqsrt.2016.08.004, 2016.

---

## Author Comment (AC1) · 25 Feb 2017

**We would like to thank the reviewer for their valuable suggestions and time. Our responses are given below.**

Anonymous Referee #1

**Referee Comment:** In this work, three different methodologies were used to calculate the contributions of absorption from BC, BrC and lensing. The authors descripted the methodologies in the Results and Discussion section (Sec. 3.3). I would suggest the three approaches discussed in the Sec. 2 (Materials and Method).

**Author Response:** We think this description is very important to our result. We decided to keep descriptions in result and discussion section.

**Referee Comment:** I would suggest providing a schematic of the instrument setup in the Materials and Method section.

**Author Response:** Schematic of instrumental setup is provided in the Material and Method section and Figure numbers are updated accordingly.

**Referee Comment:** The efficiency of the thermal removal mechanism of non-refractory component is a critical point, but in this work is poorly investigated. Incomplete removal of organics (e.g. low-volatile BrC coatings on BC surface) by the thermal denuder would influence the estimate in the relative contributions from BC, BrC and lensing, especially for approach 1. It will be helpful to estimate the removal of low-volatile component after the heating stage. If the author cannot quantify the efficiency of the thermal removal, they can make a sensitive analysis to discuss the uncertainties of relative contributions from BC, rBC and lensing due to incomplete removal of low-volatile organics.

**Author Response:** We completely agree with the referee that the efficiency of the thermal removal mechanism of non-refractory components by the thermal denuder (TD) is critical and this will influence the estimation from approach 1. In general, it is not straightforward to estimate the removal of low/extremely low volatility components by the TD unless the downstream aerosol is monitored by a combination of SP2, AMS, or SMPS instruments, which was not done in the current study. Saleh et al. 2014 (measuring aerosol from the same burns as this study) estimated about 10 % of the extremely low volatile organic compounds (ELVOCs) would not be evaporated by the TD, but this value will depend on TD specifications (tube diameter, length and flow rate through TD), which are different for our two thermal denuders. Even if the fraction of non-refractory component that is not removed by the TD is accurately estimated, accurate estimation of the influence of ELVOCs in absorption enhancement and fraction of BrC absorption is difficult because they have different absorptivity (imaginary refractive index) than that of low and semi volatile components. As mentioned in page 12 Line 20-26, we strongly believe that our TD did not remove all the organics so our absorption enhancement values estimated by approach 1 are underestimated (assuming no BrC absorption at 660 nm). By approach 1, the relative contribution from BC is over estimated while BrC and lensing are underestimated, but quantifying the actual uncertainties of relative contributions from BC, BrC, and lensing is complex due to the reasons as

discussed above. This is why we present the other approaches, which are much less dependent on the effectiveness of the thermal denuder.

**Referee Comment:** Due to incomplete removal of low-volatile organics, the Eabs_660 was underestimated and the babs_405_den was overestimated in Eq. (3), which would result in an unclear uncertainty (overestimate, underestimate or counteraction) in the fraction of absorption from BrC using approach 1. Meanwhile, the assumption of coated BC with an AAE of 1 in approach 2 led to an overestimate in the fraction of absorption from BrC. I do not know why the author could conclude that the BrC contributions derived from approach 2 was closer to reality than approach 1.

**Author Response:** Due to incomplete removal of low-volatile organics, estimated fraction of absorption due to BrC using approach 1 is most likely underestimated. The logic is that Eq. (3) can be simplified as

$$b_{abs\_405\_BrC} = b_{abs\_405\_dry} - E_{abs\_660} \times b_{abs\_405\_den}$$

$$b_{abs\_405\_BrC} = b_{abs\_405\_dry} - \frac{b_{abs\_660\_dry}}{b_{abs\_660\_den}} \times b_{abs\_405\_den}$$

The denuded absorption at both 660 and 405 nm will be overestimated due to incomplete removal of organics, but the problem is expected to be worse at 405 nm because both brown carbon and lensing increase the 405 nm denuded absorption while lensing is the dominant effect for the 660 denuded absorption. Given this, the ratio $\frac{b_{abs\_405\_den}}{b_{abs\_660\_den}}$ is expected to be larger than one and hence BrC absorption will be underestimated because both the dry absorptions (405, 660 nm) will not be affected.

[Figure]

The figure above is a combination of panels d-f from Figure 7 shows the fraction of absorption due to BrC at 405 nm vs AAE for all approaches. As depicted from the figure, approach 1 is close to approach 3 when AAE is small but approach 2 is close to approach 3 when AAE is high. This clearly shows that if aerosol is dominated by BrC, the fraction of absorption by BrC estimated by approach 1 is much smaller than that estimated by approach 2 or 3. Approach 3 is thought to be the maximum possible impact from lensing (AAE = 1.6) and therefore the fact that approach 1 is below approach 3 strongly suggests that approach 1 is low. This result may be different from other studies if TD removes non-refractory materials more efficiently. This is why we concluded approach 2 would be close to reality.

**Added Text Location:** Section 3.3, Page 10, Line 17

**Added Text:** Due to incomplete removal of low-volatile organics, estimated fraction of absorption due to BrC using approach 1 is most likely underestimated. The logic is that Eq. (3) can be simplified as

$$b_{abs\_405\_BrC} = b_{abs\_405\_dry} - E_{abs\_660} \times b_{abs\_405\_den}$$

$$b_{abs\_405\_BrC} = b_{abs\_405\_dry} - \frac{b_{abs\_660\_dry}}{b_{abs\_660\_den}} \times b_{abs\_405\_den}$$

The denuded absorption at both 660 and 405 nm will be overestimated due to incomplete removal of organics, but the problem is expected to be worse at 405 nm because both brown carbon and lensing increase the 405 nm denuded absorption while lensing is the dominant effect for the 660 denuded absorption. Given this, the ratio $\frac{b_{abs\_405\_den}}{b_{abs\_660\_den}}$ is expected to be larger than one and hence BrC absorption will be underestimated because both the dry absorptions (405, 660 nm) will not be affected.

**Referee Comment: Page 1 Line 17**: The temperature unit should be "C".
**Author Response:** The temperature unit has been changed to C.

**Referee Comment: Page 1 Line 18-20**: Three approaches were used to calculate the contributions of absorption from BC, BrC and lensing. However, the author only described two methodologies (i.e. with one: : :: : :.and the other: : :..).

**Author Response:**
    We describe the two extreme approaches in the abstract and leave the approach that lies between these two extremes for the body of the paper.

**Referee Comment: Page 7 Line 12-14**: Please define the absorption Angstrom exponent (AAE), such as using an equation.
**Author Response:** The AAE is now defined as an Equation.

**References:**

Saleh, R., Robinson, E. S., Tkacik, D. S., Ahern, A. T., Liu, S., Aiken, A. C., Sullivan, R. C., Presto, A. a, Dubey, M. K., Yokelson, R. J., Donahue, N. M. and Robinson, A. L.: Brownness of organics in aerosols from biomass burning linked to their black carbon content, Nat. Geosci., 7, 647–650, doi:10.1038/ngeo2220, 2014.

---

## Author Comment (AC2) · 25 Feb 2017

**We would like to thank the review for valuable suggestions and time. Our responses to the reviewer comments are given below.**

Anonymous Referee#2

**Referee Comment:** Several assumptions are used when estimating the contributions of BC, BrC and lensing effect to total absorption coefficients. One of the assumptions is that the absorption enhancement from lensing is constant at all wavelengths, and this assumption is used in Approach 1 and 2. Another assumption is that clear coated BC has an AAE of 1.6, and this assumption is used in approach 3. According to references introduced by authors, these two assumptions are based on Mie calculation when all BC are core-shell mixed with other aerosol components. However, the mixing state of BC in this study is unknown. Obviously, the mixing state of BC will exert significant influences on the calculated contributions of BC, BrC and lensing effect. More analysis and discussions are needed to address the possible mixing the state of BC. I suggest that the authors estimate the influences of mixing state based on calculations in this study.

**Author Response:** We agree with the reviewer that mixing state of BC will have significant influence on estimated BC, BrC, and lensing fractional absorption for approach 3 where the AAE = 1.6 assumes all BC is internally mixed. However, mixing state is implicitly included in approaches 1 and 2. In these two approaches we measure the absorption enhancement and that measurement is of the aerosol in its actual mixing state, whatever that may be. Assuming that the absorption enhancement is the same at the different wavelengths is not affected too much by mixing state, though the actual absorption enhancement clearly is. The complications of mixing state is a major reason why we compare the BC, BrC, and lensing fractional absorption from different approaches, to see how much we can parameterize with no knowledge of the mixing state, which is often not known in practice. We investigate the extremes of how the mixing state of BC will affect the calculated values by comparing the result from approach 2 and approach 3. From equation 9 absorption from BrC is calculated as:

$$b_{abs\_\lambda_1\_BrC} = b_{abs\_\lambda_1\_dry} - b_{abs\_660\_dry} \times \left(\frac{660}{\lambda_1}\right)^1 \quad \text{(Alternate form of equation 7)} \qquad (1)$$

This approach is equivalent to assuming BC is externally mixed (lensing is negligible) while approach 3 assumes the maximum effects from lensing (BC are core-shell mixed, as mention by the reviewer). On Page 11 Line 11 we have a subsection titled, "Alternate description of approach 2" that discuss this. By comparing the BrC absorption fraction from approach 2 and 3 we quantify the potential influence of mixing state of BC on calculated BrC fractional absorption.

**Referee Comment:** Results of parallel runs of thermally denuded channel and dry channel demonstrate that significant discrepancy exists (larger than 50 % which is at the outer limits of the Gaussian error curve) between measured absorption coefficients at 405 nm from two different photoacoustic absorption spectrometers. Though results of added experiments show that the dry 405 channel is better and absorption coefficient of the denuded channel is corrected according results of parallel runs, large bias still exist in measured absorption coefficient at 405 nm and this error will propagate into the calculation of AAE and influences the analytical results introduced in

this research. Hence, uncertainty analysis about calculated AAE should be given in the text to acknowledge the awareness of this problem.

**Author Response:** We agree with the reviewer that uncertainty in absorption measurement will propagate to the estimated AAE values. However, we estimated AAE from the slope of the least squares fit to the logarithm of absorption coefficient vs. the logarithm of wavelength at 405, 532, and 660 nm. As discussed in the paper, the uncertainties in absorption coefficient at 405, 532, and 660 nm are different. Given this complication, we decided to report one standard deviation of the slope as uncertainty in the AAE values.

**Referee Comment:** Title of table 1, "at 405 nm estimated from four different approaches", it should be three approaches.

**Author Response:** We changed "four different approaches" to three different approaches.

**Referee Comment:** Caption of Figure 2, the unit of absorption coefficient should be included.

**Author Response:** The units of the absorption coefficient are now reported in the caption of Figure 2.

---

## Author Comment (AC3) · 25 Feb 2017

**We would like to thank the reviewer for their valuable suggestions and time. Our responses are given below.**

Anonymous Referee #3

**Referee Comment:** For the correlation analyses between Eabs and AAE, EC/OC ratio, or MCE and between the fraction of BrC absorption and AAE, EC/OC ratio, or MCE, the authors sometimes use logarithm but not in some other cases. For example, the author reported that the log(Eabs_Den_405) linearly correlates with AAE (Fig. 4), and the "% of absorption of BrC" linearly correlates with log(AAE) (not AAE itself) (Fig. 7). In contrast, the log-log plot was used for both of Eabs vs. EC/OC and % of absorption of BrC vs. EC/OC. The authors need to add detailed explanation on these choices. For the discussion of these correlations, the term "logarithm of" should be added in the text, if the correlation analysis was conducted for the logarithm.
**Author Response:** The referee has two good observations. We replaced the term "log" by "logarithm of" in the text. We have also added the following paragraph to the text to explain the criteria by which we chose the variables for regression.

**Added Text Location:** Section 3.2, Page 9, Line 3

**Added Text:** It is notable that some regressions are done for a semi-log plot while others are linear or log-log. The type of regression was chosen based on objective criterion for simple regression. Namely that the residuals are equally scattered from the regression line and that the residuals are as close as possible to a normal distribution. The model (either LogY vs LogX, logY vs X, or Y vs LogX) which satisfied these criterion for simple linear regression was chosen.

**Referee Comment:** This paper reported the small lensing effect for all fuels and burning conditions. Is the magnitude of the lensing effect reasonable, if you assume all OC is used for the coating with the core-shell structure?
**Author Response:** $E_{Abs}$ at 660 nm, which we attribute to lensing, varied from $0.92 \pm 0.09$ to $1.43 \pm 0.17$ (Page 8 Line 24) which is consistent with the range reported in previous literature (McMeeking et al., 2014; Lack et al., 2012). As mention in the text, the average BC core and particle size during FLAME-4 was 100 and 200 nm respectively, which, based on Mie core/shell theory would give $E_{Abs}$ of roughly 1.5 assuming internal mixing and varying slightly depending on the refractive indices utilized in the analysis (McMeeking et al., 2014).

**Referee Comment:** The main findings of the paper "The fraction of absorption from BrC shows reason-ably good correlation with AAE and EC/OC at both 405 and 532 nm." are not surprising, because the relative contribution of BrC is expected to increase with increasing OC concentration and that the AAE is expected to increase with increasing BrC (OC). The results imply that the light absorbing properties (such as mass absorption cross section and imaginary part of refractive index) of OC do not largely change with EC/OC ratio, fuel types, and burning conditions. If so, the results obtained in this work may be inconsistent with the results of Saleh et al. (2014). I think that the discussion on this point and the more detailed descriptions on the evidence leading to the conclusion be-low should be added. "This result is distinct but not inconsistent with Saleh et al. (2014) who found that the imaginary index of refraction increases with increasing BC/OA ratio.

These two results can be understood with the idea that brown carbon grows darker as emissions have a higher fraction of black carbon relative to non-refractory organic mass, but the fraction of total absorption caused by brown carbon increases as the amount of organic mass increases and the black carbon to organic carbon mass ratio decreases."

**Author Response:** Perhaps the main findings that "The fraction of absorption from BrC shows reasonably good correlation with AAE and EC/OC at both 405 and 532 nm." are not surprising to the reviewer, but they are certainly notable. There are many variables that contribute to absorption including, most notably, the mixing state of the aerosol and the refractive index of the organic material, but also including the morphology of the black carbon core and other variables. Given all this, it is very interesting that a simple correlation of this type can reasonably represent the fraction of absorption by brown carbon. The reviewer is incorrect in stating that the results show that the imaginary refractive index of the organic carbon does not change with EC/OC ratio. This would be implied if there was a linear relationship between the amount of OC (not EC/OC ratio) and the absorption coefficient. The result shown, a linear correlation between the % of absorption by brown carbon and the EC/OC ratio, is far more complex and is not easily reduced to find the imaginary refractive index of the organic aerosol without knowledge of the black carbon size distribution, the total size distribution, and the mixing state of the aerosol. Because we do not have this data available, we feel that the statement currently made about our results relative to those of Saleh et al. (2014) is all that can be said. We agree with the reviewer that this is an area that needs further study to reconcile these results and we are actively involved in this research now.

**Referee Comment: Page 3, Line 2 Material and Methods:** If a part of the data (fuel) used in this work is same with that used in Pokhrel (ACP2016), it is better to add the information on that and how the authors chose the data (fuel) used in this work.

**Author Response:** Both this paper and Pokhrel et al., 2016 are from data collected during FLAME-4 and they are based on the same data set. This is already described in the paper and Pokhrel et al., 2016 is repeatedly referenced. The papers are very different. Pokhrel et al (2016) discusses the single scattering albedo and absorption angstrom exponent dependence on MCE and EC/OC. These optical properties are very different from $E_{abs}$ and the fraction of absorption from brown carbon discussed in this work. The only real overlap between the results presented in the two papers is the use of AAE values.

**Referee Comment: Page 4, 2.1 Inlet system**

    a) Have you estimate the particle losses in the Nafion dryer and the activated carbon monolith?

        **Author Response:** We do not discuss the particle losses in the Nafion dryer and the activated carbon monolith because both of these were on upstream of all PAS channels any losses will have similar effects on all channels. Since our results are all intensive properties particle losses will not affect the results unless the dramatically alter the size distribution, which is not expected because large aerosol was removed with the cyclone impactor and the vast majority of the mass was typically much smaller than a micron.

b) Did you check the possible contribution of removal of semi-volatile organic compounds to the amount and optical properties of BrC through the change in the gas/particle partitioning?
   **Author Response:** There is always concern that semi-volatiles could be lost in an inlet system, but there is no way to quantify if that occurred in the current dataset because all data was collected through the inlet. The PAS and the EC/OC instruments were on different inlets.

c) Line 17: How often did you insert the filter for the baseline measurements?
   **Author Response:** A filter was performed every 5 to 10 minutes. This statement has been added to the text.

**Referee Comment: Page 5, 2.4 Particle loss in Thermal denuder**

a) How did you measure the particle size dependence of particle loss? Did you use two SMPS system and place them upstream and downstream of thermal denuder?
   **Author Response:** A poly-disperse aerosol was measured going through the denuder and bypassing the denuder and the two size distributions were divided by one another to generate the size dependent particle losses shown.

b) In Fig. 1, the particle transmittance only above 100 nm were given. How was the transmittance for smaller particles? If the particles with a diameter less than 100 nm are negligible, I recommend to the authors to give some information on this point.
   **Author Response:** Particle transmission for particles less than 100 nm size was noisy. In terms of mass (on which absorption coefficient depends), these particles have a small contribution.

**Referee Comment: Page 6, lines 22-24** "The calibration of the dry 405 nm channel determined without the high ozone points (what was done for most of the project) was consistently closer to the slope determined using all ozone concentrations (including the high ozone points) than the calibration of the denuded 405 nm channel without the high-ozone points." => I recommend to adding more qualitative information on this point.
**Author Response:** Except for the last three days of experiments, ozone calibration of the PAS was performed at concentration levels which gives maximum absorption of 20 Mm$^{-1}$ at 405 nm. During last three days, the ozone calibration was performed at much higher concentrations of ozone and absorption at 405 nm peaked at 40 Mm$^{-1}$. For the dry 405 nm channel, the calibration constant (Slope of PAS IA vs CRDS extinction) remained fairly constant for both conditions whereas calibration constant of the denuded 405 nm channel showed significant drift. Hence we decided to adjust denuded channel to match dry channel. We feel the paragraph stated is sufficient for the paper but have added detail here.

**Referee Comment: Page 7, line 31** "For fires without backup filters or those that were below the detection limit, the average OC correction for that fuel type was applied: rice straw (2.0± 0.4 %), ponderosa pine (1.2 %), black spruce (2.9± 1.6 %), and peat (3.1± 0.8 %). For fuel types without backup filters collected, the study average OC artifact (2.4± 1.2 %) was subtracted." => It seems

that the authors assume the amount of carbonaceous gas adsorption is proportional to the mass concentration of OC. The assumption may not be reasonable if the filter is saturated.

**Author Response:** We agree with referee about the assumption that is made in applying this artifact correction. We have added a statement to this effect, as well as evidence that the filter was not saturated.
**Added Text Location:** Section 2.7, Page 8, Line 3

**Added Text:** This approach to artifact correction assumes that the amount of carbonaceous gas adsorbed is proportional to the mass concentration of OC; this assumption is considered to be reasonable because the back-up filters contained less than 5.6 $\mu$g OC cm$^{-2}$ and similar quartz fiber filters become saturated above 6 $\mu$g OC cm$^{-2}$ (Turpin et al., 1994).

**Referee Comment: Page 9, line 24** Are the particle sizes (100 and 200 nm) given here volume-based average diameters or number based average diameters?
**Author Response:** These are the number based average diameters
**Referee Comment: Page 14, line 2** "Figure 5" may be "Figure 7"
**Author Response:** We changed this to Figure 7.

References:

Turpin, B. J., Huntzicker, J. J., and Hering, S. V.: Investigation of organic aerosol sampling artifacts in the Los-Angeles basin, Atmos. Envir., 28, 3061–3071, 1994.

---

## Author Comment (AC4) · 25 Feb 2017

**We would like to thank C. He for the comment. Our response is given below.**

**Short comment by C. He:

**Short Comment:** The authors used different methods to investigate the relative contributions of BC, BrC, and coating enhancement to the total absorption of biomass burning aerosols. The analysis and results could improve our understanding on aerosol absorption for biomass burning emissions. I have a short comment.
Recent studies showed that BC optical properties are also significantly influenced by particle (coating) structures in addition to coating thickness (He et al., 2015, 2016), which could be an important uncertainty source in determining aerosol absorption. It would be useful if the author could include these references and add some discussions on this aspect to highlight potential uncertainty associated with these important factors in affecting BC/BrC absorption.

**Author Response:** We appreciate the comment by C. He and find the work cited to be relevant and interesting. However, we do not feel that it would be appropriate to insert a discussion of how advanced models that include BC morphology predict adjustments to the optical properties of coated BC because we do not present experimental results that can verify or reject the importance of these particular effects.